# SPoVT: Semantic-Prototype Variational Transformer for Dense Point Cloud Semantic Completion

**Sheng-Yu Huang**[1*]   **Hao-Yu Hsu**[1*]   **Yu-Chiang Frank Wang**[1,2]

[1]Graduate Institute of Communication Engineering, National Taiwan University

[2]NVIDIA

{f08942095, r10922195}@ntu.edu.tw, frankwang@nvidia.com

## Abstract

Point cloud completion is an active research topic for 3D vision and has been widely studied in recent years. Instead of directly predicting the missing point cloud from the partial input, we introduce a *Semantic-Prototype Variational Transformer (SPoVT)* in this work, which takes both partial point cloud and their semantic labels as the inputs for semantic point cloud object completion. By observing and attending to geometry and semantic information as input features, our SPoVT would derive point cloud features and their semantic prototypes for completion purposes. As a result, our SPoVT not only performs point cloud completion with varying resolution, it also allows manipulation of different semantic parts of an object. Experiments on benchmark datasets would quantitatively and qualitatively verify the effectiveness and practicality of our proposed model.

## 1  Introduction

3D computer vision has been a popular research topic throughout recent years, related to various extensive applications such as autonomous vehicles, augmented reality, and graphical rendering. As one of the most commonly used data representations, point clouds can be easily acquired by 3D sensors. However, these 3D scans are often incomplete and sparse due to self-occlusion or far distance from sensors, leading to undesirable results for further applications (e.g., 3D object detection for LiDAR point clouds often fails when target objects are far from the sensor). Therefore, recovering full point cloud data from partial observations becomes an important task.

As pioneers of object-level point cloud completion, Yuan *et al.* [1] propose PCN, combining a PointNet encoder [2] and a folding-based decoder [3]. Since PCN and the subsequent folding-based methods [4, 5, 6, 7, 8] only adopt shared multi-layer perceptron (MLP) for point-wise decoding, detailed geometry information might not be properly recovered. To alleviate this problem, [9, 10] voxelize the partial point cloud so that 3D CNN can be directly used for local feature propagation. To better refine local geometry, [11, 12, 13, 14] use Transformer [15]-based frameworks and apply attention between neighboring points. Nevertheless, these approaches simply rely on 3D coordinates as input features for the point cloud, and thus how to preserve the geometry and the associated semantic information for each part of the output point cloud would be a challenge.

Different from object-level completion approaches, recent scene-level completion methods [16, 17, 18, 19, 20, 21] show that point cloud completion and semantic segmentation can be jointly performed by sharing semantic and geometrical information. With additional ground truth for 3D segmentation

---

[*]Equal Contribution

Table 1: **Comparisons between different point cloud completion methods.**

| Methods | Setting | Input | | Capability | | | |
| --- | --- | --- | --- | --- | --- | --- | --- |
| | | xyz | Sem. Label | Completion | Sem. Label | Global / Part manipulation | Varying resolution |
| PoinTr [14] | Object | ✓ | - | ✓ | - | -/- | - |
| VRC-Net [12] | Object | ✓ | - | ✓ | - | ✓/ - | - |
| PCSSC-Net [20] | Scene | ✓ | ✓ | ✓ | ✓ | -/- | - |
| Ours | Object | ✓ | ✓ | ✓ | ✓ | ✓/✓ | ✓ |

labels as supervision, these approaches enforce 3D input data to capture geometries of individual objects in a scene. However, it is not clear how such techniques can be easily extended to object-level point cloud completion with semantics properly preserved or manipulated.

In this paper, we propose a Semantic-Prototype Variational Transformer (SPoVT) for semantics-preserving point cloud completion. Given a partial point cloud of an object with semantic labels observed for each point, our SPoVT is able to derive point cloud features and their semantic prototypes. With feature distributions for each semantic part properly observed, point cloud completion and semantic segmentation can be jointly achieved. Our SPoVT can be viewed as an encoder-decoder-based transformer, whose encoding process is to derive the *semantic prototypes* for each object part and their point-wise geometry features, while the decoding part is to recover the complete point cloud via sampling from the derived semantic-specific point cloud distribution. The proposed framework, as we discuss later, is not limited to the completion of point clouds at a particular resolution (i.e., varying numbers of point cloud outputs). In addition, we show that the semantic prototypes allow us not only to recover but also to manipulate object parts for diverse completion outputs.

We now summarize the contributions of this work as follows:

- We propose a Transformer-based network, Semantic-Prototype Variational Transformer (SPoVT), taking partial point cloud and the associated part labels for point cloud completion.

- By taking the above inputs, our proposed network learns point cloud distributions for each semantic part, allowing us to resample point features for decoding and to generate point clouds with varying resolutions.

- A ratio predictor is deployed in SPoVT for predicting point number distributions for each segment part, which serves and guidance for point cloud completion and alleviates potentially dense or sparse completion for particular object parts.

- By learning prototypes and feature distributions for object parts, our model is able to perform point cloud completion and manipulate at instance or part levels.

## 2 Related work

### 2.1 Object completion

With the recent development of point cloud analysis [2, 22, 23], point cloud shape completion becomes a popular topic. As a pioneer, Yuan *et al.* [1] propose PCN, a simple encoder-decoder network based on FoldingNet [3]. By concatenating global features obtained from PointNet [2] with 2D points and passing through MLPs, PCN simulates the process of "folding" a 2D plane into the 3D surface of an object. Tchapmi *et al.* [4] further introduce a tree-structured folding approach, while Wang *et al.*repeat the folding process multiple times to iteratively grow and refine their prediction in CRN [5] and SCRN [6]. On the other hand, ASFM-Net [7] adapts feature matching between latent codes obtained from the complete and the partial point clouds for improved point cloud completion. Since these folding-based approaches directly apply shared MLPs to the concatenations of global features and predefined 2D points during decoding, interactions between neighboring points are not considered, leading to possible coarse outputs without preserving local geometries.

To tackle the above issue, recent methods [10, 9, 12, 11, 13, 24, 14, 25] develop different approaches to achieve geometry-aware point cloud completion. VE-PCN [10] and GRNet [9] voxelize point cloud features via aggregating point features within voxel grids, directly apply 3D CNN for encoding and decoding and then regenerate point cloud by inverse gridding. VRCNet [12], PMP-Net++ [11] and PoinTr [14], on the other hand, introduce variants of Transformer [15] for point cloud completion.

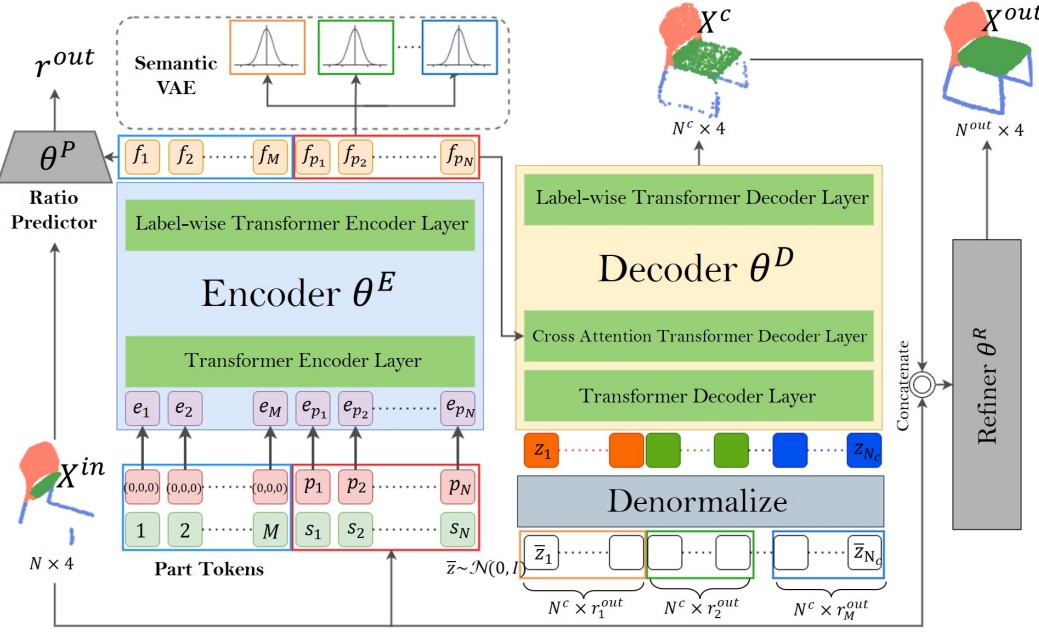

Figure 1: Architecture of our proposed Semantic-Prototype Variational Transformer (SPoVT), which contains Encoder $\theta^E$, Ratio Predictor $\theta^P$, Decoder $\theta^D$, and Refiner $\theta^R$. Note that Semantic VAE is introduced for learning prototypes and feature distributions for each object parts during completion.

In order to focus on local geometry and to reduce computation, they only consider features from neighboring points when implementing attention modules. Although very promising results have been presented, using only 3D coordinates of partial inputs and relying on derived global features might not be sufficient due to the lack of semantic cues during completion.

### 2.2 Scene completion

SSCNet [26] proposes to complete both a scene and the corresponding semantic segmentation from a single 3D scan via 3D CNN. To perform more realistic completion by taking semantics as inputs, recent approaches [16, 17, 18, 19, 20, 21] make use of additional semantic related input and show that 3D partial observations and semantic information are complementary to each other by exhibiting impressive results. For example, AIC-Net [16] combines multi-scale 2D and 3D features to make sure objects of different sizes are completed correctly. And, SPAwN [17] directly fuses voxelized 3D features with 2D priors obtained from a segmentation map to generate a detailed indoor scene.

Nevertheless, the above methods are not easily applied to complete and manipulate 3D object point clouds. This is the reason why our work focuses on taking both point cloud geometry and semantic information as inputs for deriving the associated part prototypes and distributions. In Table 1, we compare the characteristics of recent point cloud completion models with our SPoVT.

## 3 Proposed method

### 3.1 Problem formulation and model overview

For the sake of completeness, we first determine the setting and notations used in this paper. We have an input partial point cloud $X^{in}$, containing a total of $N$ points in 3D along with the associated semantic label. That is, we have $X^{in} = \{(\boldsymbol{p}_n, \boldsymbol{s}_n) \mid n = 1, 2, \ldots, N\}$, where $\boldsymbol{p}_n \in \mathbb{R}^3$ denotes the $n$-th point with its attributes describing coordinates $(x_n, y_n, z_n)$, and $\boldsymbol{s}_n$ is the corresponding semantic label (i.e., $\boldsymbol{s}_n \in \{1, 2, \ldots, M\}$, where $M$ is the total number of part classes). Our goal is to predict a complete point cloud $X^{out}$ with $N^{out}$ points, given the ground truth $X^{GT}$ with $N^{in}$ points during training.

To achieve this goal, we propose a Semantic-Prototype Variational Transformer (SPoVT) in this paper. As depicted in Fig. 1, our SPoVT has an *Encoder* $\boldsymbol{\theta}^E$ that learns the point features $\boldsymbol{f}_{\boldsymbol{p}_1:\boldsymbol{p}_N} = \{\boldsymbol{f}_{\boldsymbol{p}_1}, \ldots, \boldsymbol{f}_{\boldsymbol{p}_N}\}$ for input point cloud and the prototypes of each semantic part $\boldsymbol{f}_{1:M} = \{\boldsymbol{f}_1, \ldots, \boldsymbol{f}_M\}$. In particular, a learning scheme of *Semantic VAE* is presented during encoding, which learns point cloud distribution for each semantic part given the above features. Another unique characteristic is that a *Ratio Predictor* $\boldsymbol{\theta}^P$ is introduced for predicting the point number distribution across semantic parts, with output $\boldsymbol{r}^{out}$ as an $M$-dimensional vector. As for the decoding process, we deploy a coarse *Decoder* $\boldsymbol{\theta}^D$ and a *Refiner* $\boldsymbol{\theta}^R$, aiming to recover a coarse point cloud $X^c$ and the refined output $X^{out}$, respectively.

It is worth repeating that, by taking both partial point cloud and the associate part labels, our SPoVT is learned to jointly perform point cloud completion and semantic segmentation. Due to the ability to derive semantic prototypes for each part and to describe their feature distributions, instance/part-based point cloud synthesis and manipulation can be further achieved. We now detail our SPoVT in the following subsections.

## 3.2 Semantic-Prototype Variational Transformer

### 3.2.1 Encoding

Given the partial input $X^{in}$, our Encoder $\boldsymbol{\theta}^E$ in Figure 1 is designed to derive features for each point $\boldsymbol{f}_{\boldsymbol{p}_1:\boldsymbol{p}_N}$ and semantic prototypes for each part $\boldsymbol{f}_{1:M}$. In order to preserve both semantic and geometrical information during encoding, $\boldsymbol{\theta}^E$ is designed to take a sequential input with length $M+N$. The first $M$ inputs are viewed as the tokens for semantic prototypes, i.e., $\boldsymbol{e}_{1:M} = \{\boldsymbol{e}_1, \ldots, \boldsymbol{e}_M\}$, each $\boldsymbol{e}_m$ contains the 3D coordinates (initialized by $(0, 0, 0)$) and the semantic label $m$. The latter $N$ inputs denote the partial point cloud, i.e., $\boldsymbol{e}_{\boldsymbol{p}_1:\boldsymbol{p}_N} = \{\boldsymbol{e}_{\boldsymbol{p}_1}, \ldots, \boldsymbol{e}_{\boldsymbol{p}_N}\}$, with each $\boldsymbol{e}_{\boldsymbol{p}_n}$ described by the 4D representation of $(\boldsymbol{p}_n, \boldsymbol{s}_n)$. And, the encoded outputs would be $\boldsymbol{f}_{1:M}$ and $\boldsymbol{f}_{\boldsymbol{p}_1:\boldsymbol{p}_N}$, describing the embeddings derived for each prototype and input point cloud. It is worth noting that, instead of performing standard self-attention during encoding, we replace the last encoder layer with a *Label-wise Transformer Layer*, attending point cloud and the semantic parts with the same labels.

**Semantic VAE.** In order to better describe each semantic part and its feature distribution so that the output point cloud can be properly completed, we introduce the learning scheme of Semantic VAE in our SPoVT. Given the point features $\boldsymbol{f}_{\boldsymbol{p}_1:\boldsymbol{p}_N}$ and the semantic prototypes $\boldsymbol{f}_{1:M}$, we take the point features associated with label $m$ and map its posterior distribution $q_{\theta_m}(\boldsymbol{f}_{\boldsymbol{p}_i} \mid X^{in})$ to a predefined prior distribution $\mathcal{N}(\boldsymbol{0}, \boldsymbol{I})$. Note that each of these distributions is parameterized by $\theta_m$ via the reparameterization trick introduced in [27], and $\mathcal{N}(\boldsymbol{0}, \boldsymbol{I})$ denotes the standard normal distribution.

Thus, the objective function for this process is defined as:

$$\mathcal{L}_{KL} = \sum_{m=1}^{M} \frac{1}{N^m} \sum_{i=1}^{N^m} \boldsymbol{KL}[q_{\theta_m}(\boldsymbol{f}_{\boldsymbol{p}_i} \mid X^{in}) \| \mathcal{N}(\boldsymbol{0}, \boldsymbol{I})], \tag{1}$$

where $N^m$ is the number of points with label $m$, and $\boldsymbol{KL}[\cdot]$ denotes the KL-divergence. Note that the point feature $\boldsymbol{f}_{\boldsymbol{p}_i}$ in Eqn. 1 is normalized by $\boldsymbol{f}_m$, and we keep the same notation for simplicity. With this Semantic VAE scheme, our SPoVT is able to perform instance/part-based point cloud synthesis and manipulation, as verified later in experiments.

**Ratio predictor.** Without the guidance of semantic labels, point cloud completion might produce dense or sparse point clouds for particular parts, which would be undesirable. Given both partial point cloud and their semantic labels, we additionally introduce a *Ratio Predictor* $\boldsymbol{\theta}^P$, which takes both the derived semantic prototypes $\boldsymbol{f}_{1:M}$ and partial point cloud $X^{in}$ as inputs to predict $\boldsymbol{r}^{out}$, which as an $M$-dimensional vector indicating the point number distribution under the supervision of $X^{GT}$. Thus, the ratio loss for this ratio predictor is determined as $\mathcal{L}_{ratio} = \mathcal{D}(\boldsymbol{r}^{GT}, \boldsymbol{r}^{out})$, where $\boldsymbol{r}^{GT}$ denotes the ground truth point ratio and $\mathcal{D}(\cdot)$ is a distance function. For simplicity, here we consider L2 distance as $\mathcal{D}(\cdot)$. With this additional guidance, the decoding process can be performed accordingly.

### 3.2.2 Decoding

The decoding process of SPoVT is composed of a coarse Decoder $\boldsymbol{\theta}^D$ and a Refiner $\boldsymbol{\theta}^R$. The former outputs a coarse completion with $N^c$ points from the complete point features sampled from the learned point distributions, while the latter predicts $N^{out}$ point-wise displacements by considering both the partial input and the coarse output to refine the results.

In order to select proper inputs to $\boldsymbol{\theta}^D$, our encoder design allows us to sample point features for each semantic part with predicted point number distribution. In other words, we simply sample $N^c$ $D$-dimensional random noise vectors $\overline{\boldsymbol{z}}_{1:N^c} = \{\overline{\boldsymbol{z}}_n \mid n = 1, 2, \ldots, N^c\}$ from the Normal distribution for each part. The part labels $\boldsymbol{s}^c_{1:N^c} = \{\boldsymbol{s}^c_n \mid n = 1, 2, \ldots, N^c\}$ are assigned to each sample according to the predicted $\boldsymbol{r}^{out}$, leading to $N^c_m$ points for each label $m$, where $N^c_m = r'_m \times N^c$ and $r'_m$ denotes the $m$-th entry of $\boldsymbol{r}^{out}$. Finally, we recover these input noise vectors $\overline{\boldsymbol{z}}_{1:N^c}$ into the complete point features $\boldsymbol{z}_{1:N^c}$ by denormalizing them according to the corresponding semantic prototypes.

**Coarse completion.** The coarse Decoder $\boldsymbol{\theta}^D$ takes the recovered point features $\boldsymbol{z}_{1:N^c}$ as the inputs and predicts a coarse point cloud output $X^c$, where $X^c = \{(\boldsymbol{p}^c_n, \boldsymbol{s}^c_n) \mid n = 1, 2, \ldots, N^c\}$. Inspired by [15], we use a cross-attention layer as the second layer of $\boldsymbol{\theta}^D$ to preserve point-wise structure from the partial input, with the last layer replaced by a *Label-wise Transformer Layer* for preserving semantics. The objective for coarse completion is the L2-Chamfer Distance:

$$\mathcal{L}_{\mathcal{CD}}(X^c, X^{GT}) = \frac{1}{N^c} \sum_{i=1}^{N^c} \mathcal{CD}(\boldsymbol{p}^c_i, X^{GT}) + \frac{1}{N^{GT}} \sum_{j=1}^{N^{GT}} \mathcal{CD}(\boldsymbol{p}^{GT}_j, X^c), \tag{2}$$

where the point-to-set Chamfer Distance $\mathcal{CD}(\boldsymbol{p}^c_i, X^{GT})$ of the $i$-th coarse predict point $\boldsymbol{p}^c_i$ to the ground truth point cloud $X^{GT}$ is defined as:

$$\mathcal{CD}(\boldsymbol{p}^c_i, X^{GT}) = \min_{(\boldsymbol{p}^{GT}, \boldsymbol{s}^{GT}) \in X^{GT}} \| \boldsymbol{p}^c_i - \boldsymbol{p}^{GT} \|_2, \tag{3}$$

and the point-to-set Chamfer Distance $\mathcal{CD}(\boldsymbol{p}^{GT}_j, X^c)$ of the $j$-th ground truth point $\boldsymbol{p}^{GT}_j$ to the coarse prediction $X^c$ is defined similarly.

**Refinement.** For the task of point cloud completion, it is necessary to preserve geometrical details for the given partial point cloud. Thus, we deploy a Refiner $\boldsymbol{\theta}^R$ which takes the concatenation of partial input $X$ and coarse output $X^c$ as the inputs $X^{cat}$ for refined decoding. To be more precise, we have $X^{cat} = \{(\boldsymbol{p}^{cat}_n, \boldsymbol{s}^{cat}_n) \mid n = 1, 2, \ldots, N^{out}\}$, where $N^{out} = N + N^c$, $(\boldsymbol{p}^{cat}_n, \boldsymbol{s}^{cat}_n) = (\boldsymbol{p}_n, \boldsymbol{s}_n) \forall n \in (1, \ldots, N)$, and $(\boldsymbol{p}^{cat}_n, \boldsymbol{s}^{cat}_n) = (\boldsymbol{p}^c_n, \boldsymbol{s}^c_n) \forall n \in (N + 1, \ldots, N^{out})$. Our $\boldsymbol{\theta}^R$ then takes concatenation of the coordinates of each $\boldsymbol{p}^{cat}_n$ and the corresponding $\boldsymbol{f}_m$ as the inputs for predicting the coordinate displacement $\boldsymbol{d}_n$. As a result, our final completion result $X^{out}$ is obtained as $X^{out} = \{(\boldsymbol{p}^{out}_n, \boldsymbol{s}^{out}_n) \mid n = 1, 2, \ldots, N^{out}\}$, where $\boldsymbol{p}^{out}_n = \boldsymbol{p}^{cat}_n + \boldsymbol{d}_n, \forall n \in (1, \ldots, N^{out})$.

However, despite the above refiner, the use of standard L2 Chamfer Distance might still lead to noisy results [8]. Thus, we propose a Gated Chamfer Distance as a novel objective, which regularizes outlier output points and favors the inliers with small displacements. More specifically, the Gated Chamfer Distance is defined as:

$$\mathcal{L}_{\mathcal{GCD}}(X^{out}, X^{GT}) = \frac{1}{N^{out}} \sum_{i=1}^{N^{out}} \mathcal{L}_{pred}(\boldsymbol{p}^{out}_i, X^{GT}) + \frac{1}{N^{GT}} \sum_{j=1}^{N^{GT}} \mathcal{CD}(\boldsymbol{p}^{GT}_i, X^{out}),$$

$$\mathcal{L}_{pred}(\boldsymbol{p}^{out}_i, X^{GT}) = \begin{cases} \mathcal{CD}(\boldsymbol{p}^{out}_i, X^{GT}), & \text{if } \mathcal{CD}(\boldsymbol{p}^{cat}_i, X^{GT}) \geq T, \\ \| \boldsymbol{d}_i \|_2, & \text{otherwise,} \end{cases} \tag{4}$$

where the threshold $T$ is calculated as: $T = \frac{1}{N^{out}} \sum_{i=1}^{N^{out}} \mathcal{CD}(\boldsymbol{p}^{out}_i, X^{GT})$. As shown in (4), if the coordinate of a point $\boldsymbol{p}^{cat}_i$ before refinement is sufficiently close to the ground truth, we consider it as an inlier and suppress the magnitude of its displacement $\boldsymbol{d}_i$. Oppositely, for an outlier point, we apply the standard Chamfer Distance term to penalize it. Finally, the overall loss for the decoding process is $\mathcal{L}_{decode} = \mathcal{L}_{\mathcal{CD}} + \mathcal{L}_{\mathcal{GCD}}$.

Table 2: **Quantitative evaluation on PCN in terms of L2-Chamfer Distance (CD$\times 10^4$) and mIoU (%).** Note that $N^{GT} = 16384$ for all methods across different categories.

| Method | Airplane | | Car | | Chair | | Lamp | | Table | | Avg. | |
|---|---|---|---|---|---|---|---|---|---|---|---|---|
| | CD | mIoU | CD | mIoU | CD | mIoU | CD | mIoU | CD | mIoU | CD | mIoU |
| PCN [1] | 1.26 | 67.4 | 10.8 | 38.1 | 5.77 | 79.3 | 11.4 | 62.1 | 5.22 | 76.6 | 6.88 | 64.7 |
| PMP-Net++ [11] | 1.80 | 70.3 | 3.82 | 48.6 | 3.42 | 75.3 | 7.93 | 66.3 | 7.87 | 59.3 | 4.97 | 64.0 |
| VRC-Net [12] | 0.84 | 69.7 | 3.15 | 60.6 | 3.50 | 82.2 | 4.90 | 75.5 | 4.76 | 74.1 | 3.43 | 72.4 |
| PoinTr [14] | 1.88 | 53.6 | 3.73 | 50.8 | 3.01 | 79.2 | 4.55 | 60.5 | 2.97 | 76.1 | 3.23 | 64.0 |
| SPoVT* | 0.75 | 82.1 | 2.99 | 76.9 | 2.97 | 77.0 | 4.50 | 86.1 | 3.04 | 84.1 | 2.85 | 81.2 |
| SPoVT (Ours) | **0.73** | **82.6** | **2.86** | **82.5** | **2.36** | **85.2** | **4.12** | **91.5** | **2.50** | **86.5** | **2.51** | **85.7** |

## 3.3 Training and Inference

**Training.** Since the part labels $s^c_{1:N^c}$ for the coarse output are assigned during decoding according to $r^{out}$ without being predicted by another segmentation model, we do not need any additional objective for the segmentation results. Instead, to make sure $\theta^D$ is able to preserve the correct semantics, we pre-train $\theta^E$ and $\theta^D$ with the reconstruction branch before starting the completion task, i.e., taking $f_{p_1:p_N}$ as inputs of $\theta^D$ and predicting the reconstructed point cloud $X^r = \{(p^r_n, s^r_n) \mid n = 1, 2, \ldots, N\}$, where $s^r_n = s_n, \forall n \in (1, \ldots, N)$. The loss function for the reconstruction is defined as:

$$\mathcal{L}_{recon} = \mathcal{L}_{mse}(X^r, X^{in}) + \lambda_{\mathcal{KL}}\mathcal{L}_{\mathcal{KL}}, \tag{5}$$

where $\mathcal{L}_{mse}(X^r, X^{in})$ is the Mean Square Error between the coordinates of $X^r$ and $X^{in}$, and $\lambda_{\mathcal{KL}}$ indicates the regularization weight of KL-divergence. Finally, we jointly optimize the reconstruction branch and completion branch by summing up the reconstruction loss $\mathcal{L}_{recon}$, the ratio loss $\mathcal{L}_{ratio}$, and the decoding loss $\mathcal{L}_{decode}$.

**Inference with varying resolution.** One unique feature of our SPoVT for point cloud completion is its ability to produce output point clouds with varying resolutions. Due to the design of Semantic VAE, our SPoVT learns a complete point distribution for each semantic part, allowing us to achieve varying resolution output by repeating the decoding process multiple times depending on the target resolution. Thus, if a total number of $N^{out} \times k$ points is required, we conduct the decoding process $k$ times with noises $\bar{z}_{1:N^c}$ resampled and $N^{out}$ points generated during each time. We then concatenate all the $k$ outputs together to obtain $kN^{out}$ points in total.

Another feature of our SPoVT is its ability to manipulate point cloud at instance and part levels. Due to Semantic VAE, the produced prototype and point features across different parts or objects can be manipulated with different point number distributions. We will verify this in our experiments and confirm its effectiveness.

## 4 Experiments

### 4.1 Dataset and Implementation Details

**Dataset.** We conduct our experiments on the PCN dataset [1], following the training and testing split provided. To obtain semantic part labels for each object, we combine the PCN dataset and a part segmentation dataset ShapeNetPart [28], both as subsets of ShapeNet [29]. As a result, we have a total number of 11262 different objects with five categories (i.e., airplane, car, chair, lamp, and table), where up to 4 part categories are available for each object. Each object is sampled with 16384 points as point cloud ground truth and has corresponding partial inputs rendered from 8 different virtual camera views. Please refer to Supplementary for more details and examples.

**Implementation details.** We set both the input point number $N$ and the coarse output point number $N^c$ to 512, leading to the total output points $N^{out} = 1024$. In all our experiments, we train one model for each object category, using a single NVIDIA TESLA V100 GPU (32G) or NVIDIA RTX 3090 GPU (24G) for training, with batch size = 16, and learning rate = $10^{-4}$ using the Adam [30] optimizer with an untuned linear warmup strategy [31]. We apply cyclical annealing [32] to $\lambda_{\mathcal{KL}}$ to avoid the potential collapse problem. After two cycles, $\lambda_{\mathcal{KL}}$ is set to $10^{-5}$. We implement our model with PyTorch [33] and PyTorch3D [34] libraries. We also use the official implementation of [12, 14, 11] and the PyTorch implementation of [1] provided by [14] for comparison.

### 4.2 Semantic Point Cloud Completion

We now compare our SPoVT with a number of state-of-the-art approaches, including PCN [1], PMPNet++ [11], PoinTr [14], and VRCNet [12] in both the completion and the part segmentation tasks. We train the model for each category separately for fair comparisons. Note that since the previous methods are all designed for completion tasks only, we follow prior semantic instance completion methods such as RFD-Net [35] and RevealNet [36] when performing comparisons. That is, we adopt DGCNN [23] as the part segmentation model, which is pre-trained on the above dataset to predict the part labels for the above methods.

**Quantitative evaluation.**  We evaluate the completion and part segmentation results with the L2-Chamfer Distance and mean intersection over union (mIoU), respectively. For the L2-Chamfer Distance, we follow VRCNet [12] and set $N^{GT} = 16384$. As for the part segmentation, the mIoU evaluates the average IoU of each part type in that object category. Note that the average scores are calculated by averaging over the score of five categories. The quantitative comparisons are summarized in Table 2. It shows that the proposed SPoVT achieves the best score on both the Chamfer Distance and mIoU among all methods. This verifies the design of encoding semantic information and learning point distribution of each part, where the sampled point features indeed represent the geometry of complete parts. To verify that our SPoVT is applicable for cases where ground truth segmentation labels are not observed, we conduct an additional experiment in which segmentation labels predicted by pre-trained DGCNN are used for training SPoVT, denoted as SPoVT$^*$ in Table 2. From this table, we see that while SPoVT$^*$ is not able to achieve comparable results as the full-version does, it still performs against SOTA methods for both completion and segmentation tasks. This suggests that our proposed model is able to utilize pre-trained segmenters to assign point cloud labels for completion and segmentation purposes. Thus, the effectiveness and practicality of our proposed model can be verified.

**Qualitative evaluation.**  The qualitative completion results are shown in Figure 2. From this figure, we observe that although existing methods complete global shapes of the desired ground truth objects, such methods do not sufficiently recover geometrical details which are not presented in the partial inputs. On the other hand, our SPoVT not only preserves details from the input but also recovers the fine geometry of the missing parts. This verifies that the design of our Semantic VAE really captures the feature distribution of each semantic part, and the sampled point features are recovered properly by the semantic prototypes. Moreover, our completion results are also the most visually uniformed among others, which shows that our Ratio Predictor indeed arranges the suitable sparsity of each part by predicting $r^{out}$ precisely.

### 4.3 Further Analysis and Ablation Study

**Manipulation.**  With the design of encoding semantic prototypes, we are able to conduct part-wise manipulation to create new objects by interpolating between specific semantic prototypes and the corresponding point ratios. Selected results are shown in Figure 3. For each row, we have a pair of the source and the target objects. We then choose one part of the source object to interpolate with the same part of the target, while the rest remain the same as the source. We can observe that the selected part of each source object gradually deforms into that of the target object. Note that if the selected part is not contained in the target, as shown in the second row in Figure 3, then the corresponding part in the source object would disappear accordingly. This verifies the design of encoding semantic prototypes, which successfully preserve both semantic and geometric information of each part.

Similarly, we demonstrate the results of instance-wise interpolation in Figure 4, which shows that our SPoVT is also capable of instance-level deformation.

**Varying resolution.**  We demonstrate the capability of varying resolution in Sec. 3.3, where our SPoVT is capable of producing varying numbers of point cloud output by repeating the decoding process multiple times. To show the benefit of this property, we compare the surface reconstruction results of our SPoVT with VRCNet [12], PoinTr [14], and the ground truth point cloud in Figure 5. While VRCNet and PoinTr are restricted to the resolution of 16384 points due to relatively deterministic predictions, our SPoVT easily achieves about $300k$ points and shows more details.

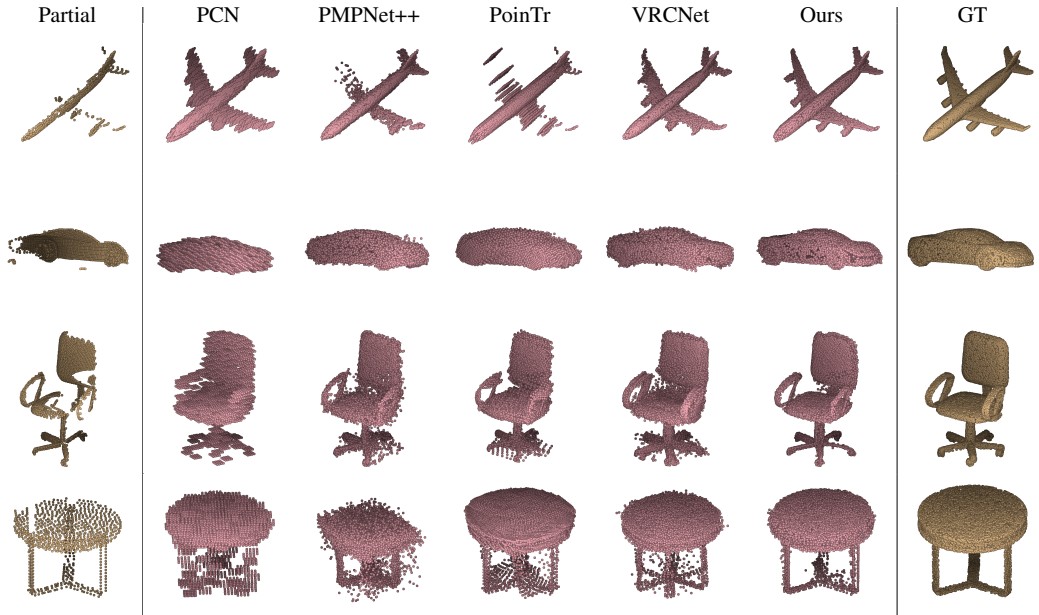

| Partial | PCN | PMPNet++ | PoinTr | VRCNet | Ours | GT |
|---------|-----|----------|--------|--------|------|-----|

Figure 2: **Qualitative evaluation.** We compare the results produced by PCN [1], PMPNet++ [11], PoinTr [14] and VRCNet [12]. Note GT represents the ground truth point cloud with $N^{GT} = 16384$.

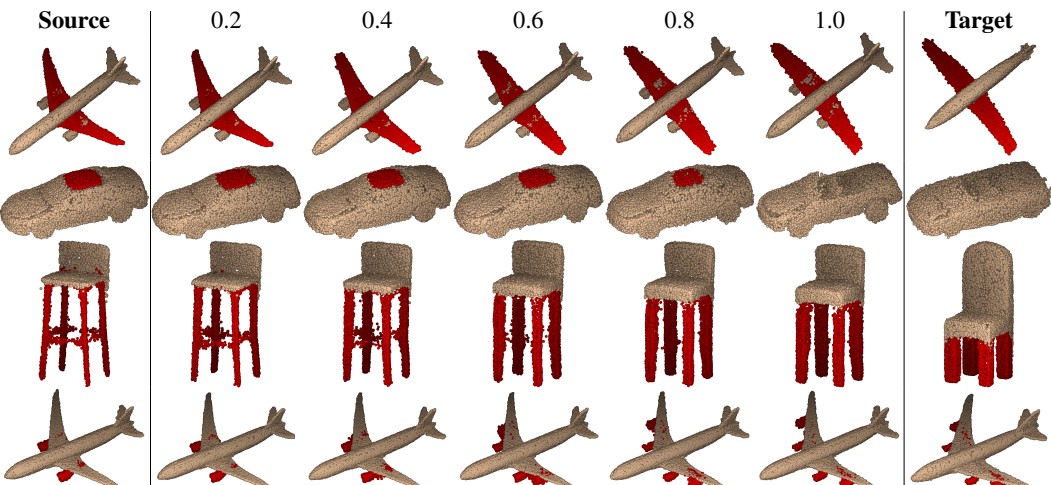

| **Source** | 0.2 | 0.4 | 0.6 | 0.8 | 1.0 | **Target** |
|------------|-----|-----|-----|-----|-----|------------|

Figure 3: **Part-level point cloud manipulation.** For each row, we select one semantic part (in red) of the source object to interpolate with the same part of the target, with the interpolation step as $0.2$.

**Ablation study.** To further analyze the effectiveness of our designed modules, learning schemes, and loss functions, we now conduct ablation studies on chair objects in Table 3. The baseline model A is composed only of the $\boldsymbol{\theta}^E$, $\boldsymbol{\theta}^D$, and $\boldsymbol{\theta}^R$ in Figure 1 without applying the reconstruction pre-train introduced in Sec. 3.3. We then conduct the reconstruction pre-train (B) to help preserve semantic information. Next, by adding back $\boldsymbol{\theta}^P$ (C), we correctly arrange the point number of each part to further raise the mIoU by $18.5\%$. In model D, benefitting from finding the point distribution of each part, our completion result improves with a $32\%$ lower Chamfer Distance. Finally, our full model (model E) using $\mathcal{L}_{\mathcal{GCD}}$ achieves the best result. Thus, the proposed modules, learning schemes, and loss function can be successfully verified.

**Point cloud completion across object categories.** In the above experiments, we train SPoVT to perform point cloud completion for each object category. It is worth noting that, one can also train SPoVT across multiple object categories by increasing the total number of semantic parts $M$. As

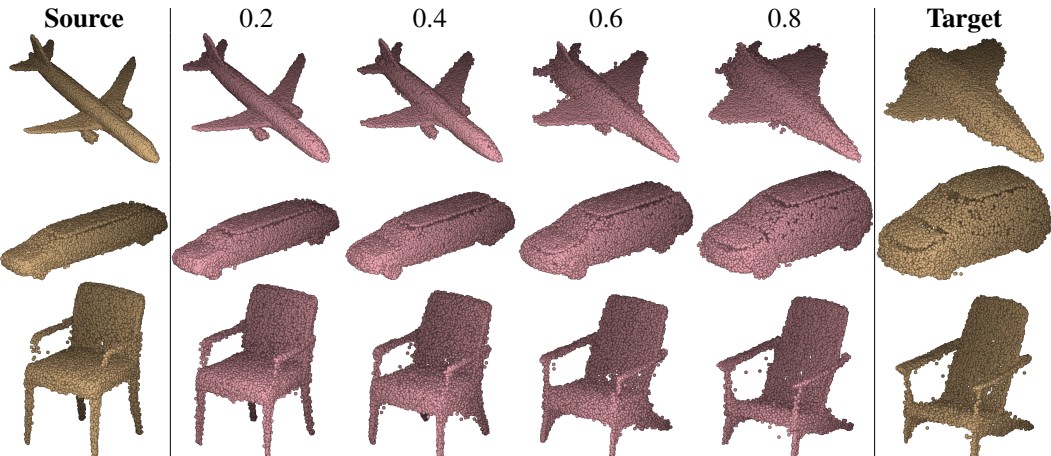

Figure 4: **Instance-level interpolation.** Given the source and target objects, we perform instance-level interpolation by interpolating between the corresponding semantic prototypes of the two objects, with the interpolation step as 0.2.

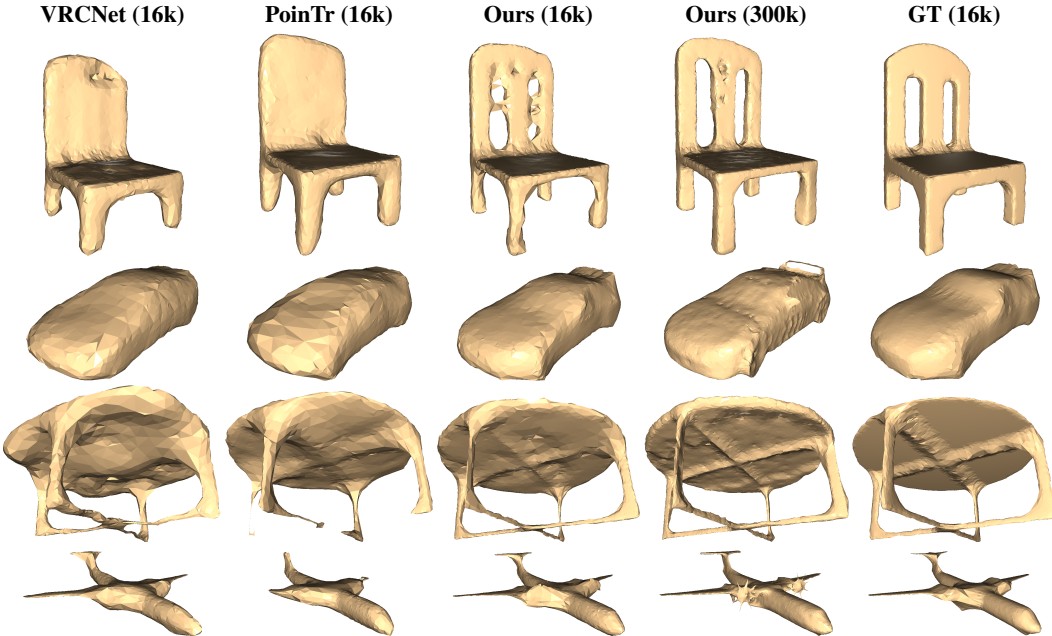

Figure 5: **Surface reconstruction with varying resolution.** The completed point cloud outputs (with numbers shown in each parenthesis) are converted to mesh outputs using the Alpha Shape method in Open3D [37]. Note that GT denotes the mesh obtained from the ground truth point cloud.

shown in Table 4, we conduct a new experiment, in which we train a single SPoVT model on both "Airplane" and "Car" categories. From the results listed in this table, we observe that a unified SPoVT exhibits slightly degraded completion and segmentation performances when compared to the original SPoVT, which is expected. Nevertheless, both SPoVT models still performs favorably against SOTA methods listed in Table 2, in which SOTA methods are trained for each object category.

## 5 Limitations

We need to point out the potential concern of applying SPoVT when performing high-resolution point cloud completion. During the inference stage, the computation time grows linearly with the output point cloud resolution, since the completion output is produced by repeating the inference process

Table 3: **Ablation studies on SPoVT.** Note the the category of *Chair* is utilized for evaluation.

| Model | Reconstruction Pretrain | $\theta^P$ | Semantic VAE | $\mathcal{L}_{\mathcal{GCD}}$ | CD | mIoU |
|---|---|---|---|---|---|---|
| A | | | | | 5.23 | 43.07 |
| B | ✓ | | | | 5.01 | 62.83 |
| C | ✓ | ✓ | | | 4.40 | 81.33 |
| D | ✓ | ✓ | ✓ | | 2.99 | 84.81 |
| E | ✓ | ✓ | ✓ | ✓ | **2.36** | **85.22** |

Table 4: **Comparison of unified trained model and original results.** We show the results that our SPoVT is trained on both "Airplane" and "Car" categories together in a unified model.

| Method | Airplane | | Car | |
|---|---|---|---|---|
| | CD | mIoU | CD | mIoU |
| Ours (Original) | 0.73 | 82.6 | 2.86 | 82.5 |
| Ours (Unified) | 0.84 | 84.6 | 3.39 | 80.6 |

Table 5: **Inference time and memory usage with different output point cloud resolution.**

| Number of points | Inference time (ms) | Memory usage (GB) |
|---|---|---|
| 2048 | 50.0 | 1.923 |
| 8196 | 145.6 | 1.923 |
| 16384 | 277.2 | 1.923 |

multiple times. In Table 5, we list the inference time and memory usage for different point cloud resolutions. Another potential issue is the slight degradation when training one single SPoVT for performing completion across multiple categories, as discussed in Sect. 4. The above are potential challenges, which are among the research topics for us to pursue in the future.

## 6 Conclusion

In this paper, we proposed a Transformer-based network of Semantic-Prototype Variational Transformer (SPoVT), which can be applied for semantic point cloud completion. By taking partial point cloud and their semantic part labels as the inputs, our SPoVT is able to derive point cloud features and the associated semantic prototypes. With the deployed Semantic VAE scheme, the point feature distributions for each semantic part can be jointly observed, which allows resampling of point features from each part for completion and manipulation. With the proposed architecture, our SPoVT is able to perform completion across resolution, with the ability to preserve the point number distribution across different semantic parts, which alleviates possible dense or sparse completion for each object part. From the experiments, we quantitatively and qualitatively confirm that our SPoVT performs favorably against state-of-the-art models in point cloud completion. We also verify its use for point cloud interpolation and manipulation between different object instances or their parts, which cannot be easily handled by most existing methods.

**Acknowledgement** This work is supported in part by the Tron Future Tech Inc. and National Science and Technology Council via NSTC-110-2634-F-002-052. We also thank National Center for High-performance Computing (NCHC) for providing computational and storage resources.

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
