# Supplementary Material of SPoVT: Semantic-Prototype Variational Transformer for Dense Point Cloud Semantic Completion

**Sheng-Yu Huang**[1][*]    **Hao-Yu Hsu**[1][*]    **Yu-Chiang Frank Wang**[1,2]

[1]Graduate Institute of Communication Engineering, National Taiwan University

[2]NVIDIA

{f08942095, r10922195}@ntu.edu.tw, frankwang@nvidia.com

## A  Dataset details and extension to real-world data

**Details of PCN dataset.**   Since all our experiments (e.g., semantic completion, surface reconstruction, global/part-wise manipulation) are evaluated on the PCN dataset [1], we now provide more details with some examples of this dataset. As mentioned in Sec. 4.1 of our main paper, we combine the PCN completion dataset and the ShapeNetPart [2] segmentation dataset to acquire part labels for both the partial input point clouds and the complete ground truth point clouds. To be more specific, we extract the intersection objects of both datasets and match the ground truth points by normalizing each pair of the same object from both datasets with the corresponding bounding box. Then we label both the partial point clouds and the complete point clouds of each object in the PCN dataset by taking the semantic label of the nearest point of the same object in the ShapeNetPart dataset. We visualize some examples of the parsed PCN dataset in Figure 1, showing that we indeed obtain correct part labels for both partial inputs and complete ground truths.

**Further extension.**   To show the capability of our SPoVT on real-world data, we follow PoinTr [3] to test the model trained with the "Car" category of PCN dataset on the KITTI dataset [4], which contains incomplete point clouds of cars in real-world scenes captured from LiDAR sensors. Since these point clouds are original without semantic labels, we pre-train a DGCNN [5] segmentation model for partial point clouds of cars in the PCN dataset. Similarly, we also test the "Chair" and the "Table" models on chairs and tables extracted from the ScanNet [6] dataset. Our comparison with PoinTr is shown in Figure 2 and Figure 3. We can see that our results better preserve the details from the partial inputs, while PoinTr gives relatively sparse results.

## B  Architecture of Refiner $\theta^R$

As described in Sec. 3.2.2, our Refiner $\theta^R$ takes the concatenation of the point coordinates of $X^{in}$ and $X^c$ as input, along with the corresponding Semantic Prototype $\boldsymbol{f}_m$ of each point. The architecture of $\theta^R$ is depicted in Figure 4, which is similar to the coarse Decoder $\theta^D$.

## C  Semantic point cloud completion and their point number distributions.

Qualitative comparisons of part segmentation are visualized in Figure 5, which shows that our SPoVT correctly completes each part of the point clouds. This verifies the design of obtaining part prototypes

---

[*]Equal Contribution

36th Conference on Neural Information Processing Systems (NeurIPS 2022).

Table 1: **Evaluation of point number distributions in predicted point clouds.** We compare the L2-distance between $r^{out}$ and $r^{GT}$, while $r^{out}$ are produced by different methods. Note that the $r^{out}$ of other methods are obtained from part segmentation results predicted by DGCNN [5].

| Method | Airplane | Car | Chair | Lamp | Table | Avg. |
|---|---|---|---|---|---|---|
| PCN [1] | 0.0640 | 0.1365 | 0.1580 | 0.3660 | 0.1772 | 0.1803 |
| PMP-Net++ [7] | 0.0902 | 0.0419 | 0.0847 | 0.0572 | 0.0549 | 0.0658 |
| VRC-Net [8] | 0.1097 | 0.1660 | 0.1117 | 0.1918 | 0.1699 | 0.1498 |
| PoinTr [3] | 0.2513 | 0.0371 | 0.1233 | 0.5187 | 0.1722 | 0.2205 |
| Ours | **0.0195** | **0.0093** | **0.0238** | **0.0044** | **0.0127** | **0.0139** |

$f_{1:M}$ in Sec. 3.2.1 in our main paper that aims to preserve the semantic information of each part. We further show the quantitative comparisons of our predicted point number distribution $r^{out}$ with PCN [1], PMP-Net++ [7], VRC-Net [8], and PoinTr [3] by evaluating the L2-distance with the ground truth point number distribution $r^{GT}$ in Table 1. With more accurate $r^{out}$, our predictions are more visually uniform on each semantic part than the others.

## D  Calculation of mIoU

Since the predicted completion $X^{out}$ and the ground truth $X^{GT}$ of a point cloud object are two different point sets, we are not able to directly apply traditional mIoU for evaluation (as part segmentation tasks [5, 9] do). Instead, inspired by Chamfer Distance, we calculate the mIoU by averaging IoU of each point in $X^{out}$ with the nearest point in $X^{GT}$ and IoU of each point in $X^{GT}$ with the nearest point in $X^{out}$. This evaluation is applied as the mIoU shown in Table 2 of our main paper.

## E  More results on point cloud completion, surface reconstruction, and part-wise manipulation.

We now provide more qualitative visualization results on point cloud completion, surface reconstruction, and part-wise manipulation in Figure 6, Figure 7, and Figure 8, respectively. Note that for the surface reconstruction in Figure 7, we directly convert the point clouds into meshes using the Alpha Shape method in Open3D [10]. For the point clouds with resolutions of 16384, such as VRCNet, PoinTr, Ours(16k), and GT, the Alpha values that decide the fineness of reconstructed mesh (the smaller, the finer) are chosen between $0.03$ and $0.04$ to make sure that the surface reconstruction is not broken. On the other hand, the Alpha value can be chosen as $0.01$ for our results with $300k$ points and show more details without creating shattered surfaces. This verifies the importance of having the property of varying resolutions that is able to predict dense point clouds.

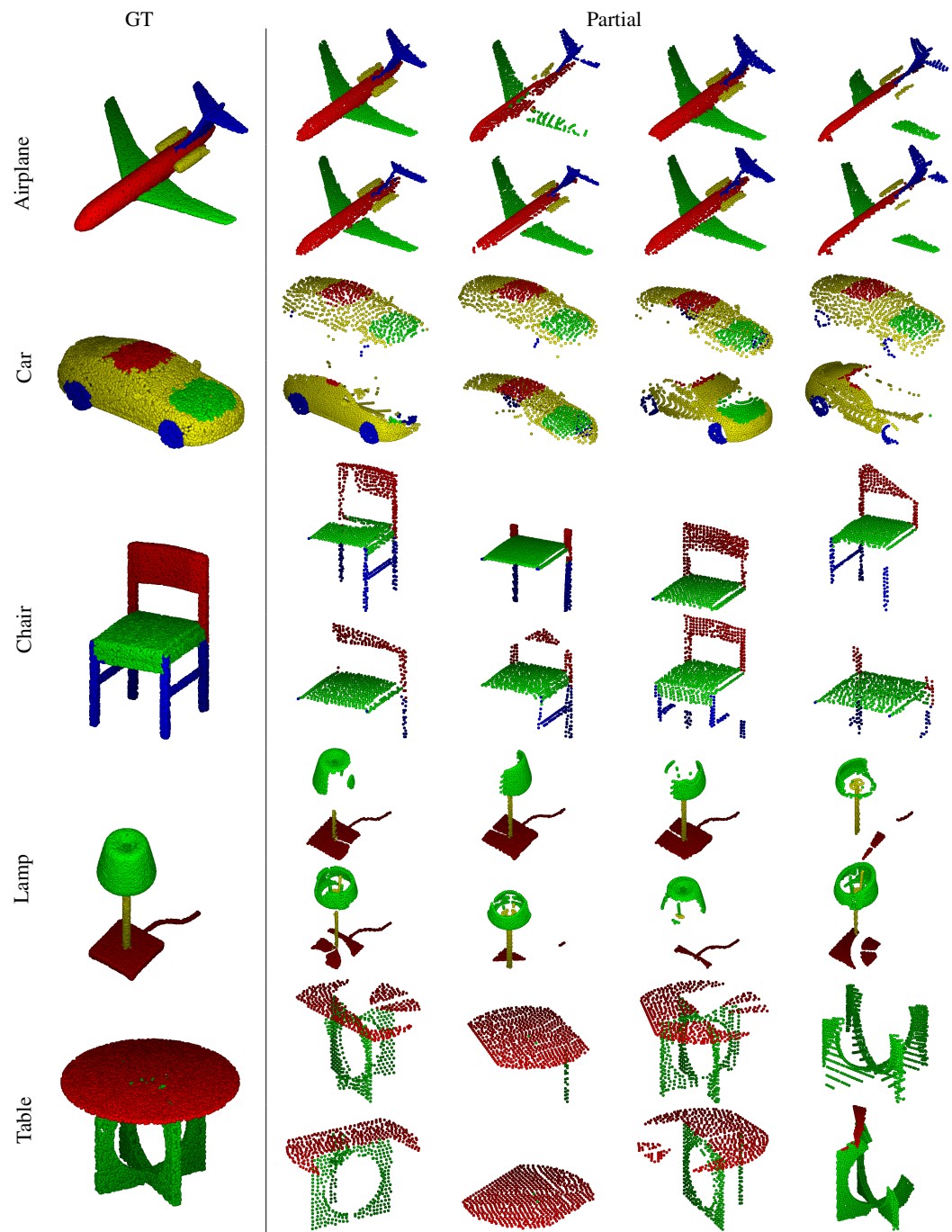

Figure 1: **Samples from the PCN dataset with segmentation labels.**

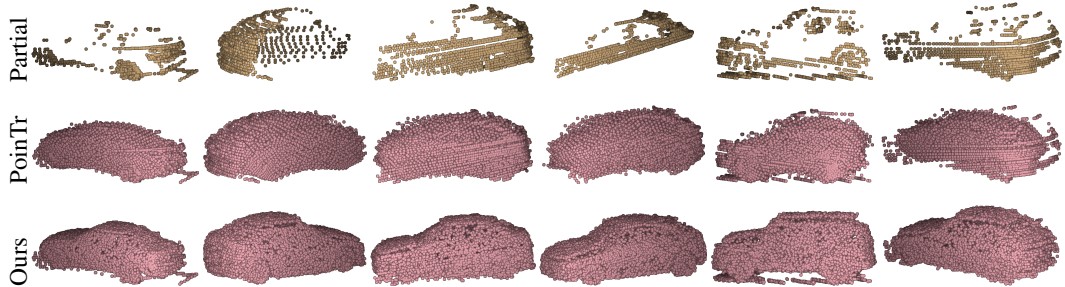

Figure 2: **Qualitative results on the KITTI dataset.** We compare our method with PoinTr [3]. Note that we pre-train a DGCNN [5] segmentation model for generating the part labels for the partial input.

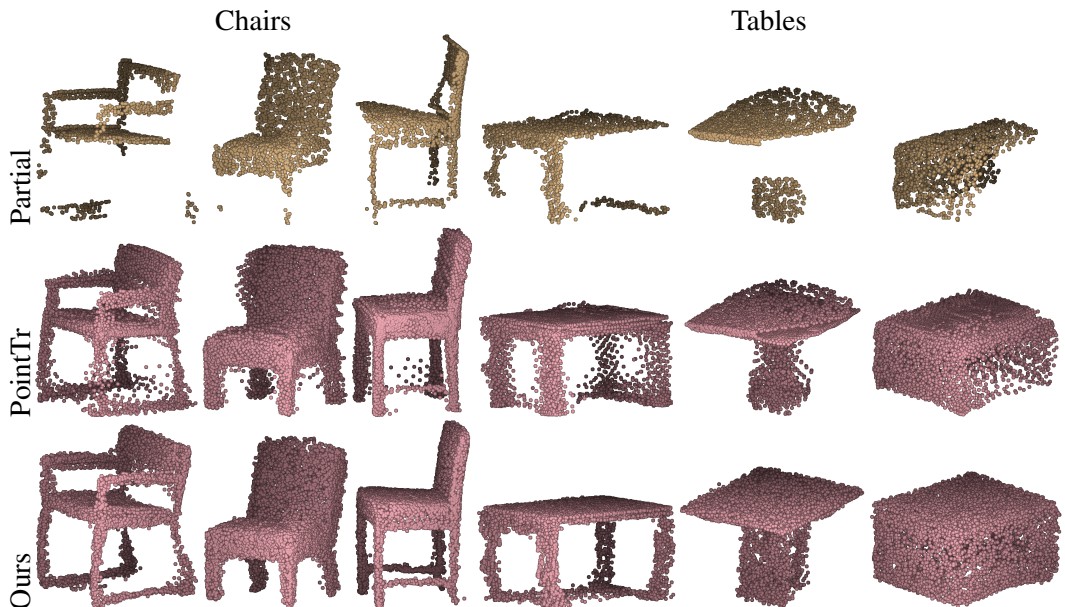

Figure 3: **Qualitative results on the ScanNet dataset.** We compare our method with PoinTr [3]. Note that the first three columns are chairs and the last three columns are tables.

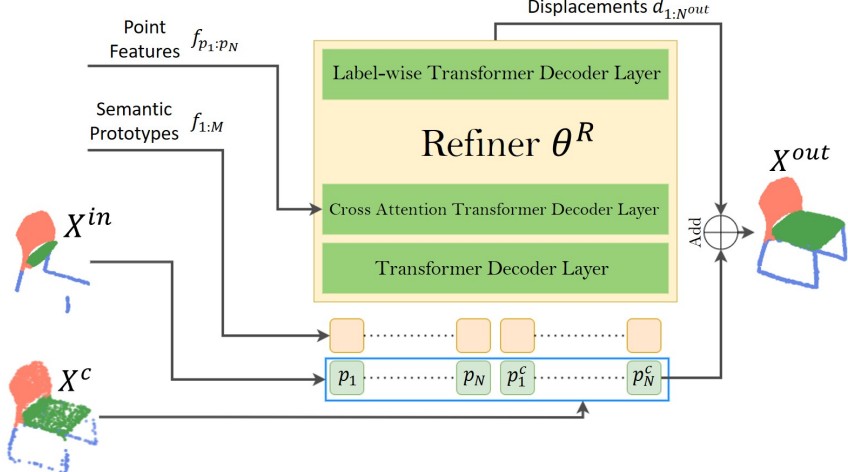

Figure 4: **Architecture of our proposed Refiner $\theta^R$.**

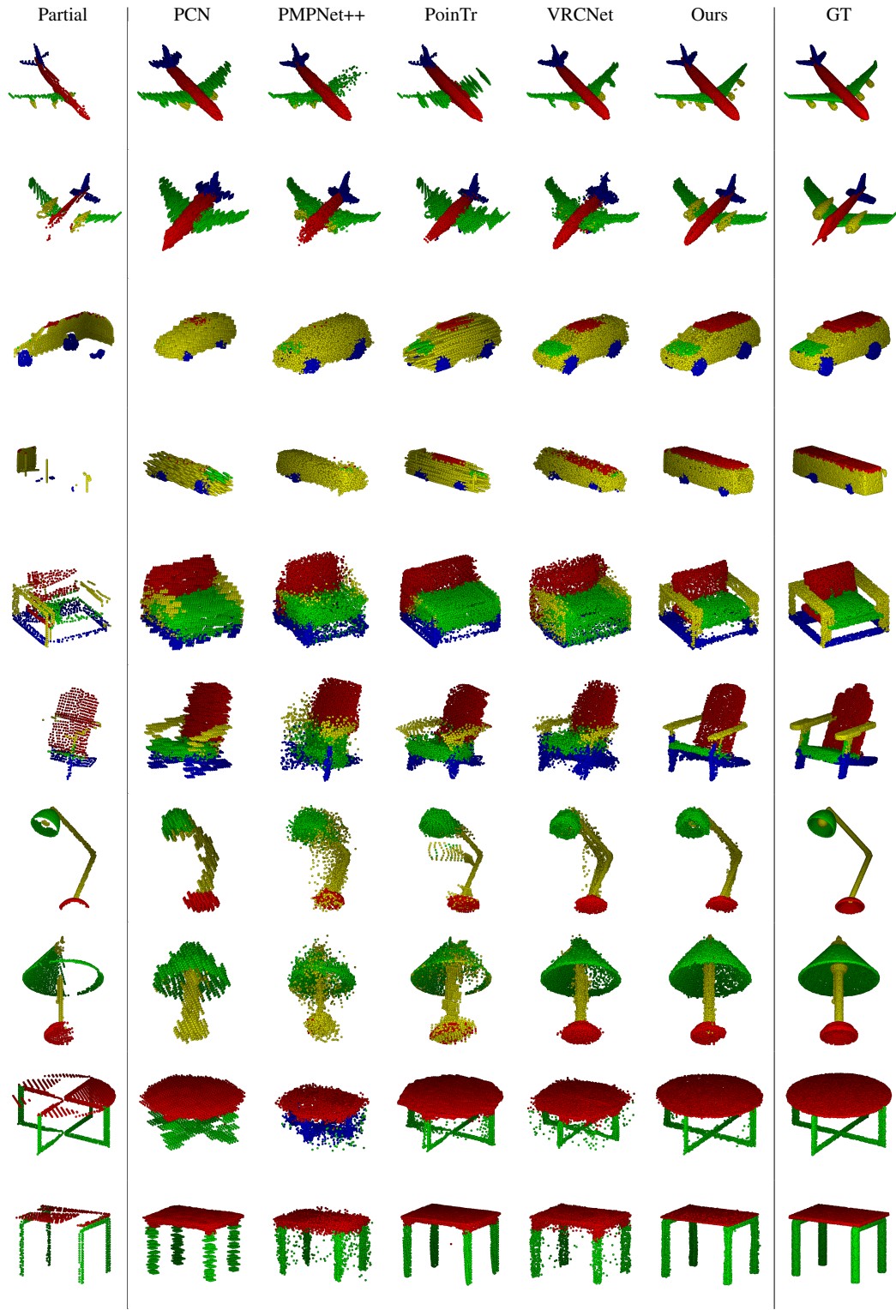

Figure 5: **Qualitative results of completed point cloud with predicted part labels.** We compare the results produced by PCN [1], PMPNet++ [7], PoinTr [3] and VRCNet [8]. Note GT represents the ground truth point cloud with $N^{GT} = 16384$. The color of each point denotes the corresponding part label.

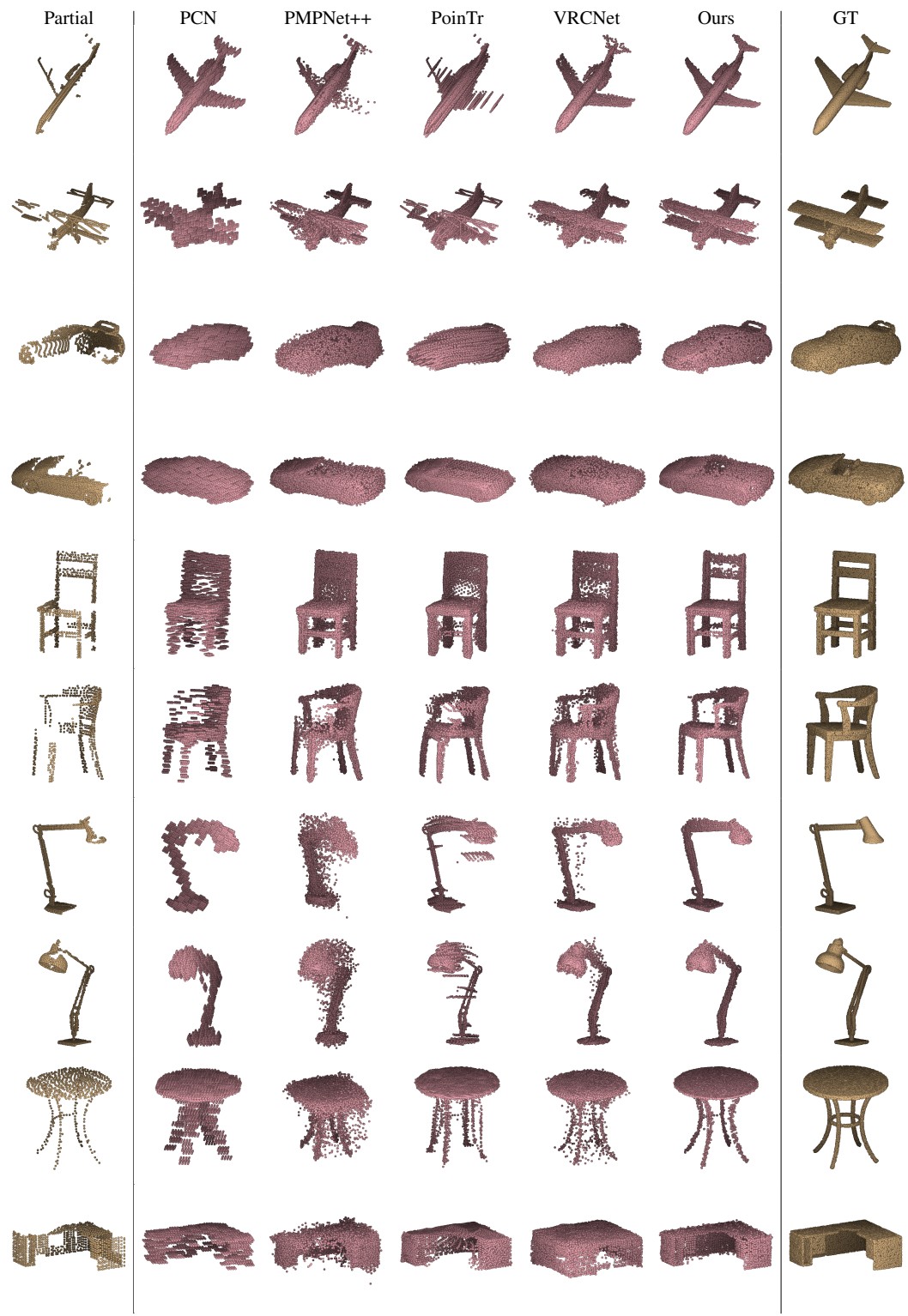

Figure 6: **Qualitative evaluation of completed point cloud.** We compare the results produced by PCN [1], PMPNet++ [7], PoinTr [3] and VRCNet [8]. Note GT represents the ground truth point cloud with $N^{GT} = 16384$.

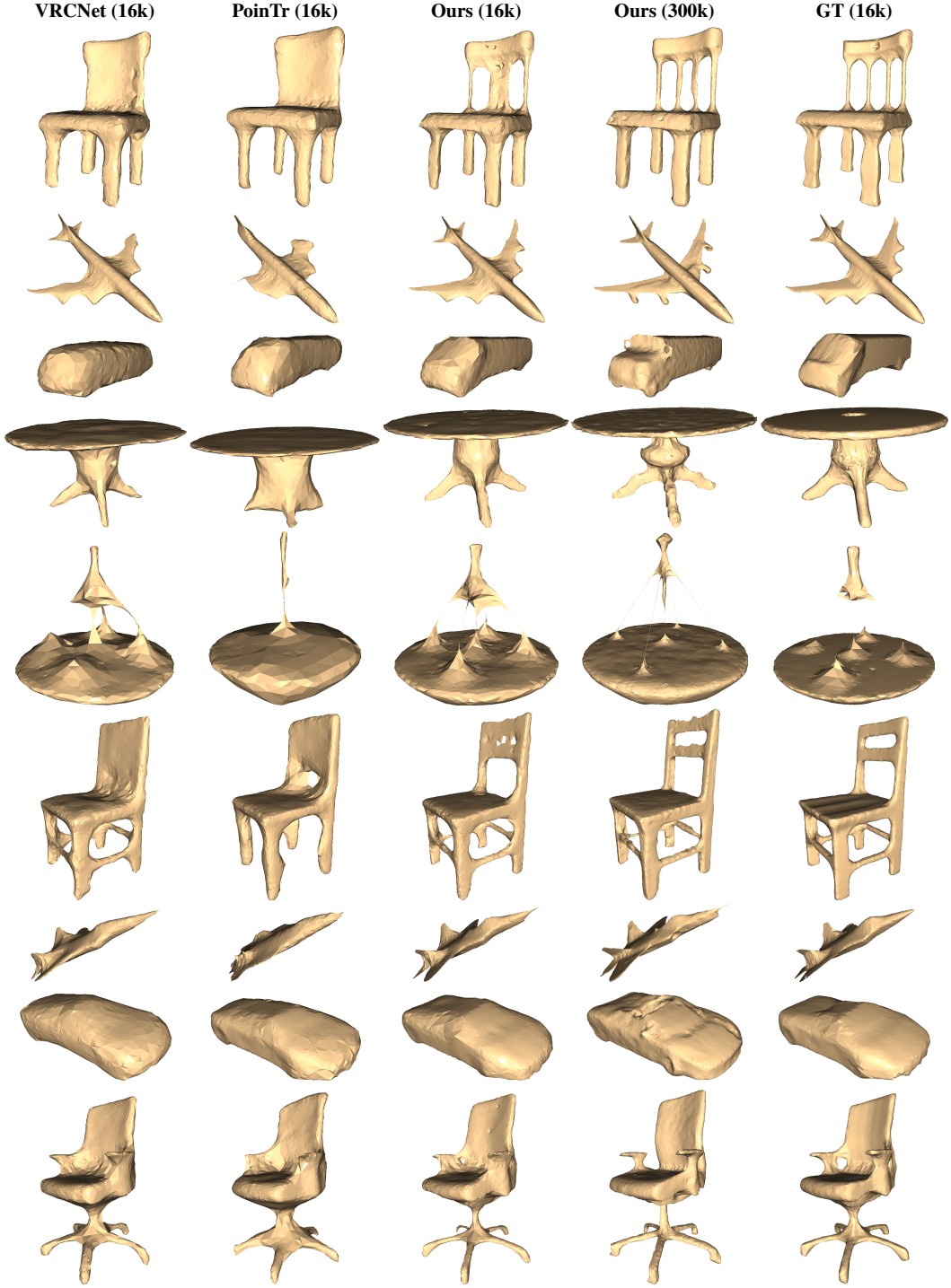

Figure 7: **Surface reconstruction with varying resolutions.** Note that GT denotes the mesh obtained from the ground truth point cloud.

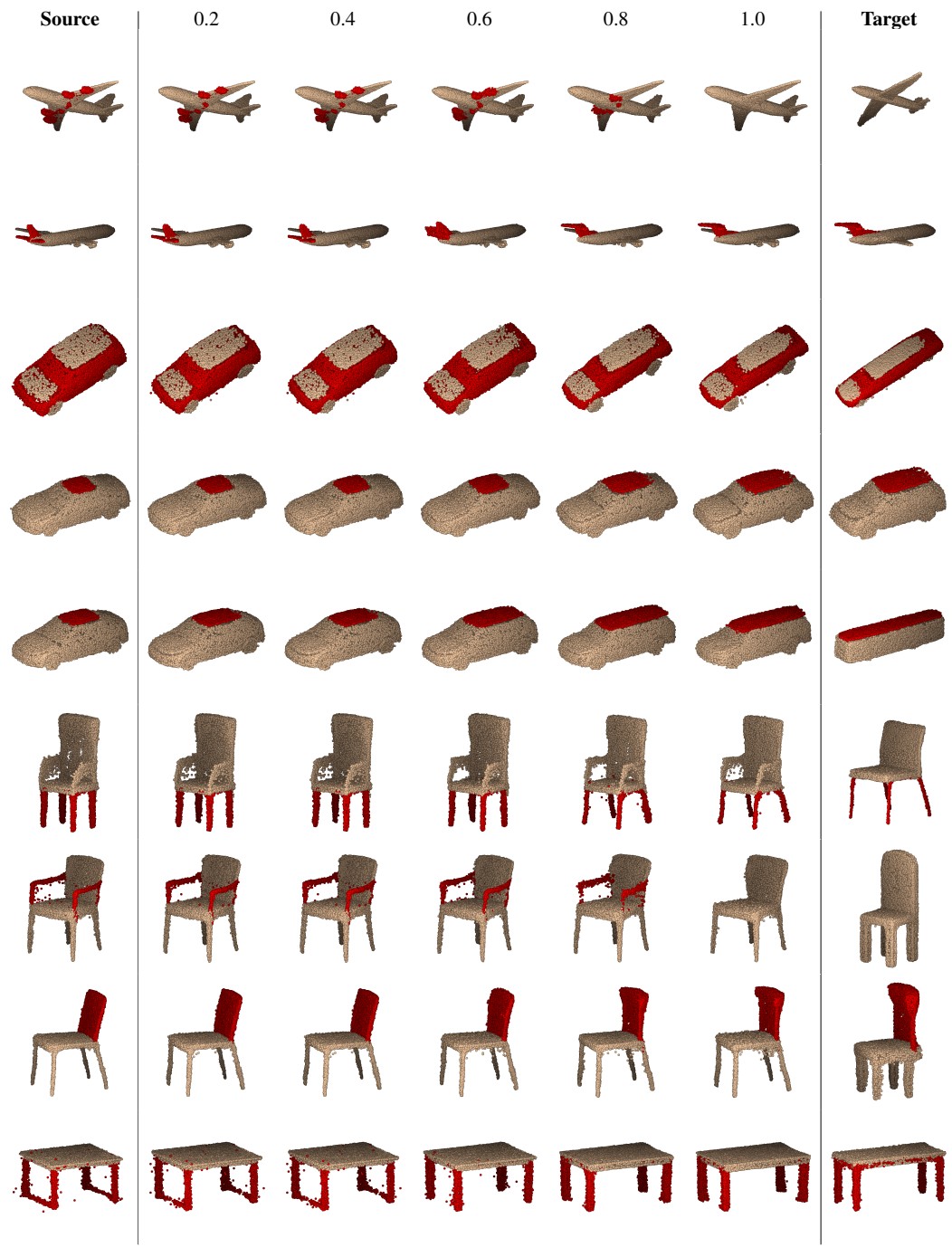

Figure 8: **Point cloud completion with part-level manipulation.** For each row, we select one semantic part (in red) of the source object to interpolate with the same part of the target, with the interpolation step as 0.2.