# OpenReview forum: "SPoVT: Semantic-Prototype Variational Transformer for Dense Point Cloud Semantic Completion"
_NeurIPS.cc/2022/Conference — NeurIPS 2022 Accept_

### Official Review · Reviewer_QsJ3 · 2022-07-10

**Rating:** 6
**Confidence:** 5
**Soundness:** 2 fair
**Presentation:** 2 fair
**Contribution:** 2 fair

**Summary:**

The authors propose Semantic-Prototype Variational Transformer for semantic point cloud completion, which combines the tasks of point cloud completion and part segmentation. The experimental results on the five categories of the PCN dataset demonstrate that the proposed method outperforms the state-of-the-arts methods.

**Questions:**

Please refer to the weaknesses.

**Limitations:**

Compared to the STOA completion methods (e.g. VRC-Net), the improvement of completion is small but requires more labor for data labeling. Moreover, the proposed method requires a model for each category. Do we really need to introduce the semantic information in point cloud completion?

**Strengths And Weaknesses:**

**Strengths**

- The paper is clearly written and well organized.

**Weaknesses**

- The motivation for introducing semantic labels in the point cloud completion is not clear. In other words, the proposed method can be regarded as point cloud completion + part segmentation.
- The experiments are only conducted on the five categories of the PCN dataset, which is not enough. PCN is a very old dataset. I suggest conducting experiments on the ShapeNet34/55 and MVP datasets.
- All competitive methods use DGCNN for part segmentation for a fair comparison. However, it is still not fair because the proposed method does not use DGCNN. Moreover, DGCNN is a very old baseline.
- Why the mIoU of PointTr are much worse than the rest methods? All of these methods adopt DGCNN for part segmentation.
- The experimental results are only compared on the synthetic dataset, how about the results on real-world scenarios?

---

> ### Author Response · Authors · 2022-08-02
> **Response to Reviewer QsJ (Part 1/3)**
>
> We thank Reviewer QsJ3 for the critical comments and suggestive remarks. Please see our responses below for each raised issue.
>
> **Q1: The motivation for introducing semantic labels in the point cloud completion is not clear. In other words, the proposed method can be regarded as point cloud completion + part segmentation.**
>
> A1: We sincerely thank the reviewer for giving us the opportunity to clarify/strengthen the motivation of our work. As pointed out in [16, 17], semantic labels have been exploited for scene-level point cloud completion. While such information is shown to be complementary to the task of completion (L29-34 and Sec. 2.2 in our main paper), most existing works for object point cloud completion do not utilize such information as the input. This motivates us to exploit point cloud labels into the task of object point cloud completion, with additional objectives for object reconstruction and part segmentation. Thus, we agree that our work can be viewed as performing joint point cloud completion and part segmentation.
>
> We further note that, existing works on object point cloud completion are not able to perform part-based point cloud manipulation, since they do not have the ability to model particular object parts during their learning process. Nevertheless, we understand the concern that observing point cloud labels might not always be practical. As later discussed in Q3, we conduct additional experiments in which with semantic labels are obtained from a pre-trained semantic labeler (i.e., DGCNN), not the ground truth ones. From the detailed results later presented and discussed in Q3, the practicality of our SPoVT can be verified.
>
>
>
>
> **Q2: The experiments are only conducted on the five categories of the PCN dataset, which is not enough. PCN is a very old dataset. I suggest conducting experiments on the ShapeNet34/55 and MVP datasets.**
>
> A2: We thank the reviewer for the constructive suggestions, and we are more than happy to conduct additional experiments. As requested, we now consider the categories of ‘Car’ and ‘Guitar’ from ShapeNet55, and we compare the outputs to those produced by PoinTr. The completion results (CD x $10^4$) are listed below:
>
> |        |  Car | Guitar | CD-Simple (avg.) | CD-Medium (avg.) | CD-Hard (avg.) |
> |--------|:----:|:------:|:----------------:|:----------------:|:--------------:|
> | PoinTr | 8.11 |  2.46  |       3.79       |       5.21       |      7.51      |
> | Ours   | 4.61 |  2.34  |       1.96       |       3.15       |      5.31      |
>
> Note that we directly apply the official implementation of PoinTr (https://github.com/yuxumin/PoinTr ), and the results are listed in the above table. Following the evaluation procedure, performances are evaluated in three difficulty categories: Simple (CD-Simple), Medium (CD-Medium), and Hard (CD-Hard), where the partial point clouds are cropping off 25%, 50%, and 75% from the ground truth point clouds. From the above table, it can be seen that our proposed SPoVT performed favorably against PoinTr across all categories. This confirms the effectiveness of our proposed model on the suggested dataset.

---

> > ### Author Response · Authors · 2022-08-02
> > **Response to Reviewer QsJ (Part 2/3)**
> >
> > **Q3: All competitive methods use DGCNN for part segmentation for a fair comparison. However, it is still not fair because the proposed method does not use DGCNN. Moreover, DGCNN is a very old baseline.**
> >
> > A3: We thank the reviewer for raising this concern. We understand that, with the use of ground truth segmentation labels for our proposed SPoVT and the use of those produced by pre-trained DGCNN for the SOTAs, the comparison in Table 2 would be less informative. And, ground truth segmentation labels might not always be available during training.
> >
> > To address and alleviate this issue, we conduct an additional experiment, in which segmentation labels predicted by pre-trained DGCNN are used for training our SPoVT (denoted as Ours* in the updated Table 2, as listed below). From the results shown in Table 2, we see that while Ours* degraded the performance when compared to the original version (Ours), it still performed against SOTA methods for both completion and segmentation tasks. This suggests that our proposed model is able to utilize pre-trained segmenters for assigning point cloud labels for completion/segmentation purposes. Thus, the effectiveness and practicality of our proposed model can be verified.
> >
> > Table 2: Quantitative evaluation on PCN in terms of L2-Chamfer Distance (CD×$10^4$) and mIOU (%). Note that $N^{GT} = 16384$ for all methods across different categories.
> >
> > | Method          | Airplane |      |  Car |      | Chair |      | Lamp |      | Table |      | Avg. |      |
> > |-----------------|:--------:|:----:|:----:|:----:|:-----:|:----:|:----:|:----:|:-----:|:----:|:----:|:----:|
> > |                 |    CD    | mIoU |  CD  | mIoU |   CD  | mIoU |  CD  | mIoU |   CD  | mIoU |  CD  | mIoU |
> > | PCN             |   1.26   | 67.4 | 10.8 | 38.1 |  5.77 | 79.3 | 11.4 | 62.1 |  5.22 | 76.6 | 6.88 | 64.7 |
> > | PMP-Net++       |   1.80   | 70.3 | 3.82 | 48.6 |  3.42 | 75.3 | 7.93 | 66.3 |  7.87 | 59.3 | 4.97 | 64.0 |
> > | VRC-Net         |   0.84   | 69.7 | 3.15 | 60.6 |  3.50 | 82.2 | 4.90 | 75.5 |  4.76 | 74.1 | 3.43 | 72.4 |
> > | PoinTr          |   1.88   | 53.6 | 3.73 | 50.8 |  3.01 | 79.2 | 4.55 | 60.5 |  2.97 | 76.1 | 3.23 | 64.0 |
> > | Ours*           |   0.75   | 82.1 | 2.99 | 76.9 |  2.97 | 77.0 | 4.50 | 86.1 |  3.04 | 84.1 | 2.85 | 81.2 |
> > | Ours (Original) |   0.73   | 82.6 | 2.86 | 82.5 |  2.36 | 85.2 | 4.12 | 91.5 |  2.50 | 86.5 | 2.51 | 85.7 |
> >
> > **Q4: Why the mIoU of PointTr are much worse than the rest methods? All of these methods adopt DGCNN for part segmentation.**
> >
> > A4: We thank the reviewer for pointing this out, and we are glad to provide an additional explanation. As shown in Fig. 5 in our supplementary material, PoinTr generally fails to complete part details (e.g., wings of the airplane). With Chamfer Distance as the only training objective, PoinTr tends to produce outputs whose shape is similar to the ground truth one. PoinTr is not designed to handle or preserve particular parts (as ours does). This explains why PoinTr was not able to reach satisfactory mIoU scores.
> >
> >
> > **Q5: The experimental results are only compared on the synthetic dataset, how about the results on real-world scenarios?**
> >
> > A5: Please kindly refer to L13-20, Fig. 2, and Fig. 3 in our supplementary material. We presented experiments on KITTI cars, ScanNet tables, and ScanNet chairs, which are all real-world datasets. With pre-trained DGCNN for predicting labels for the partial inputs (no ground truth labels available for such real-world point cloud data), we see that our model is able to produce impressive completion results and performs favorably against the SOTA ones.

---

> > > ### Author Response · Authors · 2022-08-02
> > > **Response to Reviewer QsJ (Part 3/3)**
> > >
> > > **Q6: Compared to the SOTA completion methods (e.g. VRC-Net), the improvement of completion is small but requires more labor for data labeling. Moreover, the proposed method requires a model for each category. Do we really need to introduce the semantic information in point cloud completion?**
> > >
> > > A6: We thank the reviewer for pointing out these two issues and we are glad to clarify these two issues.
> > >
> > > For the requirement for ground truth semantic labels during training, as discussed in Q3, we are able to alleviate this limitation by utilizing a pre-trained semantic labeler (i.e. DGCNN) for assigning point cloud labels. With promising and favorable results presented in Q3, the practicality of our proposed work can be confirmed.
> > >
> > > On the other hand, Reviewer Bq58 also points out the concern of training one model for each object category. In fact, as stated in L97 of our main paper, we suggest that this can be achieved by increasing the total number of semantic parts M. To better address this issue, we conduct additional experiments to assess whether our SPoVT can be trained across object categories. More specifically, we conduct a new experiment that our SpoVT is trained on both “Airplane” and “Car” categories together in a unified model. The results are shown below: (CD: x$10^4$, mIoU: %)
> > >
> > >
> > > |                 | Airplane |      |  Car |      |
> > > |-----------------|:--------:|:----:|:----:|:----:|
> > > |                 |    CD    | mIoU |  CD  | mIoU |
> > > | Ours (Original) |   0.73   | 82.6 | 2.86 | 82.5 |
> > > | Ours (Unified)  |   0.84   | 84.6 | 3.39 | 80.6 |
> > >
> > > From the above table, we observe that a unified SPoVT (trained for two object categories) exhibited slightly degraded completion and segmentation performances when compared to the original SPoVT, which is expected. Nevertheless, both models still performed favorably against SOTA methods listed in Table 2 (in which SOTA methods are trained for each category). We will be glad to add the above experiments and discussions to our revised version.

---

> > > > ### Comment · Reviewer_QsJ3 · 2022-08-06
> > > > **Final Decision**
> > > >
> > > > The authors addressed all of my concerns in the rebuttal, and I decide to change my rating to weak accept.

---

### Official Review · Reviewer_Z3zB · 2022-07-10

**Rating:** 6
**Confidence:** 3
**Soundness:** 3 good
**Presentation:** 3 good
**Contribution:** 2 fair

**Summary:**

The authors proposed a Semantic-Prototype Variational Transformer approach for the dense point cloud semantic completion problem. In addition to the partial input, the paper also used the semantic labels as input for the completion task. An encoder-decoder Transformer architecture is used to leverage both the pixel level input and the semantic label input, so that the semantic label can guide the completion task. The Variational AutoEncoder (VAE) is used for model training and inference. Empirical results showed that the semantic label helped to improve the completion quality, outperforming other baselines.

**Questions:**

It has been reported in the literature that the VAE inference is prone to posterior collapse, which greatly degrades the quality of model inference. Have the authors encountered such problem in the VAE inference in this paper? Any procedures taken to alleviate this problem (e.g., through weight annealing)?



**Limitations:**

The model uses semantic labels as additional input for point cloud completion. A natural question is, what if such additional input is not available in practice? Is it practical to run a separate semantic labeler on the point cloud?

**Strengths And Weaknesses:**

Strengths:
1) The paper proposed to use semantic label from other sources as additional input to the point cloud completion problem. With this extra source of information, the model is able to infer the ground truth point cloud with better accuracy.
2) The paper leveraged an encoder-decoder Transformer architecture to consume both the pixel level input and the semantic label input, so that the semantic label information can effectively guide the point cloud completion process.

Weaknesses:
1) The comparisons in Table 2 showed the advantages of the proposed method. However, it is not clear to me whether the comparison is apples-to-apples. For example, is the semantic label information used in other baselines (such as PCN)? It would be nice to explicitly mention which methods used such additional input, and which methods did not. That way, we would know where the improvement comes from (from the semantic label, or from the model architecture).

---

> ### Author Response · Authors · 2022-08-02
> **Response to Reviewer Z3zB**
>
>
> We thank Reviewer Z3zB for the positive comments and suggestive remarks. Please see our responses below for each raised issue.
>
> **Q1: The comparisons in Table 2 showed the advantages of the proposed method. However, it is not clear to me whether the comparison is apples-to-apples. For example, is the semantic label information used in other baselines (such as PCN)? It would be nice to explicitly mention which methods used such additional input, and which methods did not. That way, we would know where the improvement comes from (from the semantic label, or from the model architecture).**
>
> A1: We thank the reviewer for raising this critical issue. We understand that, with the use of ground truth segmentation labels for our proposed SPoVT and the use of those produced by pre-trained DGCNN for the SOTAs, the comparison in Table 2 would be less informative. And, ground truth segmentation labels might not always be available during training.
>
> To address and alleviate this issue, we conduct an additional experiment, in which segmentation labels predicted by pre-trained DGCNN are used for training our SPoVT (denoted as Ours* in the updated Table 2, as listed below). From the results shown in Table 2, we see that while Ours* degraded the performance when compared to the original version (Ours), it still performed against SOTA methods for both completion and segmentation tasks. This suggests that our proposed model is able to utilize pre-trained segmenters for assigning point cloud labels for completion/segmentation purposes. Thus, the effectiveness and practicality of our proposed model can be verified.
>
> Table 2: Quantitative evaluation on PCN in terms of L2-Chamfer Distance (CD×$10^4$) and mIOU (%). Note that $N^{GT} = 16384$ for all methods across different categories.
>
> | Method          | Airplane |      |  Car |      | Chair |      | Lamp |      | Table |      | Avg. |      |
> |-----------------|:--------:|:----:|:----:|:----:|:-----:|:----:|:----:|:----:|:-----:|:----:|:----:|:----:|
> |                 |    CD    | mIoU |  CD  | mIoU |   CD  | mIoU |  CD  | mIoU |   CD  | mIoU |  CD  | mIoU |
> | PCN             |   1.26   | 67.4 | 10.8 | 38.1 |  5.77 | 79.3 | 11.4 | 62.1 |  5.22 | 76.6 | 6.88 | 64.7 |
> | PMP-Net++       |   1.80   | 70.3 | 3.82 | 48.6 |  3.42 | 75.3 | 7.93 | 66.3 |  7.87 | 59.3 | 4.97 | 64.0 |
> | VRC-Net         |   0.84   | 69.7 | 3.15 | 60.6 |  3.50 | 82.2 | 4.90 | 75.5 |  4.76 | 74.1 | 3.43 | 72.4 |
> | PoinTr          |   1.88   | 53.6 | 3.73 | 50.8 |  3.01 | 79.2 | 4.55 | 60.5 |  2.97 | 76.1 | 3.23 | 64.0 |
> | Ours*           |   0.75   | 82.1 | 2.99 | 76.9 |  2.97 | 77.0 | 4.50 | 86.1 |  3.04 | 84.1 | 2.85 | 81.2 |
> | Ours (Original) |   0.73   | 82.6 | 2.86 | 82.5 |  2.36 | 85.2 | 4.12 | 91.5 |  2.50 | 86.5 | 2.51 | 85.7 |
>
> **Q2: It has been reported in the literature that the VAE inference is prone to posterior collapse, which greatly degrades the quality of model inference. Have the authors encountered such problem in the VAE inference in this paper? Any procedures taken to alleviate this problem (e.g., through weight annealing)?**
>
> Yes, we apply cyclical annealing [A] to the weight of KL-divergence loss to avoid the potential posterior collapse problem (see L185-190 in our main paper). After two cycles, the weight of KL-divergence loss is set to 1e-5 as stated in L218 in our main paper. We thank the reviewer for raising this issue and allowing us to clarify it. We will be happy to include the above remarks in our revised version.
>
> [A]  Cyclical Annealing Schedule: A Simple Approach to Mitigating KL Vanishing (NAACL 2019)
>
>
>
>
>
>
> **Q3: The model uses semantic labels as additional input for point cloud completion. A natural question is, what if such additional input is not available in practice? Is it practical to run a separate semantic labeler on the point cloud?**
>
> A3: We thank the reviewer for giving us this chance to clarify this issue. As we explained and confirmed in Q1, we follow the suggestion and take DGCNN as the pre-trained segmenter (i.e., semantic labeler) for all methods. This removes the requirement of observing ground truth labels during training for our proposed method. In Sect. A of our supplementary material, we run a separate semantic labeler for the real-world point cloud data to obtain their segmentation labels. This is to verify that, since ground truth labels are not available for real-world point cloud data, utilizing an additional segmentor for producing point cloud labels would be a feasible solution. As confirmed by our experiments, our model is shown to produce satisfactory completion results on such real-world data.

---

> > ### Comment · Reviewer_Z3zB · 2022-08-07
> > **Final comments**
> >
> > Thanks the authors for responding to my concerns.
> > My rating for this submission remains the same (6: Weak Accept).

---

### Official Review · Reviewer_fQ4M · 2022-07-11

**Rating:** 6
**Confidence:** 3
**Soundness:** 3 good
**Presentation:** 3 good
**Contribution:** 3 good

**Summary:**

This paper proposes the Semantic-Prototype Variational Transformer model for the 3D point cloud semantic completion problem.  Experiments on benchmark datasets show that the proposed method outperforms existing methods.

**Questions:**

From Line 259 on Page 8, the authors perform an evaluation on the setting of varying resolution, I’m curious about the memory usage and runtime comparisons.

**Limitations:**

It seems there’s a lack of evaluation on the model runtime in the paper and supplemental. It’s suggested to provide it for objective evaluation.

**Strengths And Weaknesses:**

Strengths:
The proposed approach applies transformer-like structures with a semantic VAE module for 3D point cloud semantic completion problem. The authors also apply a coarse-to-fine strategy for improving the semantic completion, with the learning prototypes and feature distributions for each object part. The ablation analysis validates the efficacy of the proposed network structure.

Weaknesses:
The motivation of using Transformer-structure for the encoding part for point cloud semantic completion seems not sufficient. Although the authors mentioned in the Introduction section that the reason is to better refine local geometry, in fact, point-based convolution methods such as PointConv [1] provide such utility and graph-based methods such as DGCNN [2] are also suitable for such task. A detailed discussion and comparisons in Related Works section are desired.

[1] Wu, Wenxuan, Zhongang Qi, and Li Fuxin. "Pointconv: Deep convolutional networks on 3d point clouds." Proceedings of the IEEE/CVF Conference on Computer Vision and Pattern Recognition. 2019.
[2] Wang, Yue, et al. "Dynamic graph cnn for learning on point clouds." Acm Transactions On Graphics (tog) 38.5 (2019): 1-12.

---

> ### Author Response · Authors · 2022-08-02
> **Response to Reviewer fQ4M**
>
> We thank Reviewer fQ4M for the positive comments and suggestive remarks. Please see our responses below for each raised issue.
>
> **Q1: The motivation of using Transformer-structure for the encoding part for point cloud semantic completion seems not sufficient. Although the authors mentioned in the Introduction section that the reason is to better refine local geometry, in fact, point-based convolution methods such as PointConv provide such utility and graph-based methods such as DGCNN are also suitable for such task. A detailed discussion and comparisons in Related Works section are desired.**
>
> A1: We greatly appreciate the reviewer for giving us the opportunity to further clarify our motivation, especially on the use of Transformer-based architectures as our network backbone.
>
> Our SPoVT is designed to derive prototypes for each semantic part, which guides the point cloud completion with the associated semantic information. The Transformer architecture allows us to assign and learn such part tokens, and thus association (i.e., self-attention) of point cloud features within/across object parts can be performed accordingly. Moreover, the above self-attention mechanism further allows us to observe and preserve geometry features for each semantic part during both encoding and decoding processes. The above remarks can be seen in L102-103 and L116-123.
>
> As for recent methods like [1, 11, 12, 14], either point-based or graph-based convolution are utilized for point cloud completion. They generally apply pooling layers to aggregate the features during encoding, which might not be able to preserve fine-grained or detailed features from the input point cloud data. It is worth repeating that, as shown in Fig. 2 and Table 2, our SPoVT is shown to preserve more part details during completion and performs favorably against SOTA methods on both completion and segmentation tasks. And, we have ablation studies in Table 3 to verify the model design of our SPoVT.
>
>
> **Q2: From Line 259 on Page 8, the authors perform an evaluation on the setting of varying resolution, I’m curious about the memory usage and runtime comparisons. It seems there’s a lack of evaluation on the model runtime in the paper and supplemental. It’s suggested to provide it for objective evaluation.**
>
> A2: We thank the reviewer for pointing this out, and we are glad to clarify this issue. As stated in L195-201, since we produce such results by repeating the inference process multiple times, the inference time only grows linearly with the point cloud resolution (but not the memory usage). Please see the table below, in which we conduct extra experiments on varying point cloud resolutions and list the required inference times and memory requirements.
>
>
> | Output point cloud resolution | Inference time (ms) | Memory usage (GB) |
> |:-----------------------------:|:-------------------:|:-----------------:|
> | 2048 points                   |         50.0        |       1.923       |
> | 8192 points                   |        145.6        |       1.923       |
> | 16384 points                  |        277.2        |       1.923       |

---

### Official Review · Reviewer_Bq58 · 2022-07-11

**Rating:** 5
**Confidence:** 4
**Soundness:** 3 good
**Presentation:** 2 fair
**Contribution:** 2 fair

**Summary:**

A Transformer-based network for point cloud completion called Semantic-Prototype Variational Transformer (SPoVT) is proposed. Firstly, SPoVT learns point cloud distributions for each semantic part by taking partial point cloud and additional associated part labels as inputs, allowing resampling point features for decoding and generation point clouds with varying resolutions. Secondly, A ratio predictor is deployed in SPoVT for predicting point number distributions for each segment part, which serves and guidance for point cloud completion and alleviates potentially dense or sparse completion for particular object parts. Finally, by learning prototypes and feature distributions for object parts, SPoVT is able to perform point cloud completion and manipulate at instance or part levels.

**Questions:**

Questions:
1.	Please address the above Weakness 1.
2.	Is the dataset the complete PCN dataset, or the intersection of PCN and ShapeNetPart? Does the intersection contain the annotations of all GT objects in the PCN?
3.	I noticed that you reproduced other methods for fair comparisons. I wonder if these methods are trained under the default configuration of the official implementation? Besides, both PMP-Net and PoinTr provide pre-trained models on PCN, how do these compare to your trained models?
4.	Just to be sure, like other completion methods, SPoVT needs to train the model separately for each category of objects, and you can't mix parts of different categories of objects, right?

Suggestions:
1.	Some suggestions for figures.
a)	Figure 1 is too cluttered, lacks annotations, and has a different style for each block. This is confusing.
b)	In Figure 3, It might be better to mark the number of points in red.
2.	The writing of the paper needs to be improved a little bit.
a)	Some paragraphs are a little cluttered, like the Semantic VAE part, ablation studies part, etc.
b)	Typos, like “Tranformer-based” in the Introduction, “SWiVT” on Table 2, “mIOU”s , etc.


**Limitations:**

No.
Suggestions: Please consider adding a discussion section to the paper to describe the limitations of the method as possible, such as the need for additional semantic annotations, inference speed, complex training process, etc.


**Strengths And Weaknesses:**

Strengths:
1.	The authors found an efficient way to introduce semantic label information in point cloud completion. Unsurprisingly, the performance is better than methods that only use partial point clouds as input.
2.	I like the idea of Semantic VAE, which can manipulate parts or objects with different point count distributions, making point cloud generation more flexible.
3.	The experimental content is sufficient.

Weaknesses:
1.	In Table 2, pre-trained DGCNN is used for semantic segmentation for each completion method for comparison. Under this condition, the comparison of the segmentation metric is not too meaningful, it just means that the segmentation of those methods + DGCNN is worse than the proposed method.
2.	Some writing problems. Please see the suggestions below.

---

> ### Author Response · Authors · 2022-08-02
> **Response to Reviewer Bq58 (Part 1/2)**
>
> We thank Reviewer Bq58 for the constructive comments and suggestive remarks. Please see our responses below for each raised issue.
>
> **Q1: In Table 2, pre-trained DGCNN is used for semantic segmentation for each completion method for comparison. Under this condition, the comparison of the segmentation metric is not too meaningful, it just means that the segmentation of those methods + DGCNN is worse than the proposed method.**
>
> A1: We thank the reviewer for raising this concern. We understand that, with the use of ground truth segmentation labels for our proposed SPoVT and the use of those produced by pre-trained DGCNN for the SOTAs, the comparison in Table 2 would be less informative. And, ground truth segmentation labels might not always be available during training.
>
> To address and alleviate this issue, we conduct an additional experiment, in which segmentation labels predicted by pre-trained DGCNN are used for training our SPoVT (denoted as Ours* in the updated Table 2, as listed below). From the results shown in Table 2, we see that while Ours* degraded the performance when compared to the original version (Ours), it still performed against SOTA methods for both completion and segmentation tasks. This suggests that our proposed model is able to utilize pre-trained segmenters for assigning point cloud labels for completion/segmentation purposes. Thus, the effectiveness and practicality of our proposed model can be verified.
>
> Table 2: Quantitative evaluation on PCN in terms of L2-Chamfer Distance (CD×$10^4$) and mIOU (%). Note that $N^{GT} = 16384$ for all methods across different categories.
>
> | Method          | Airplane |      |  Car |      | Chair |      | Lamp |      | Table |      | Avg. |      |
> |-----------------|:--------:|:----:|:----:|:----:|:-----:|:----:|:----:|:----:|:-----:|:----:|:----:|:----:|
> |                 |    CD    | mIoU |  CD  | mIoU |   CD  | mIoU |  CD  | mIoU |   CD  | mIoU |  CD  | mIoU |
> | PCN             |   1.26   | 67.4 | 10.8 | 38.1 |  5.77 | 79.3 | 11.4 | 62.1 |  5.22 | 76.6 | 6.88 | 64.7 |
> | PMP-Net++       |   1.80   | 70.3 | 3.82 | 48.6 |  3.42 | 75.3 | 7.93 | 66.3 |  7.87 | 59.3 | 4.97 | 64.0 |
> | VRC-Net         |   0.84   | 69.7 | 3.15 | 60.6 |  3.50 | 82.2 | 4.90 | 75.5 |  4.76 | 74.1 | 3.43 | 72.4 |
> | PoinTr          |   1.88   | 53.6 | 3.73 | 50.8 |  3.01 | 79.2 | 4.55 | 60.5 |  2.97 | 76.1 | 3.23 | 64.0 |
> | Ours*           |   0.75   | 82.1 | 2.99 | 76.9 |  2.97 | 77.0 | 4.50 | 86.1 |  3.04 | 84.1 | 2.85 | 81.2 |
> | Ours (Original) |   0.73   | 82.6 | 2.86 | 82.5 |  2.36 | 85.2 | 4.12 | 91.5 |  2.50 | 86.5 | 2.51 | 85.7 |
>
>
>
> **Q2: Is the dataset the complete PCN dataset, or the intersection of PCN and ShapeNetPart? Does the intersection contain the annotations of all GT objects in the PCN?**
>
>
> A2: We thank the reviewer for giving us the opportunity to clarify the dataset used in our experiments. The PCN dataset [1] is composed of 30,974 point cloud object instances across 8 categories. For each object, 8 partial observations are obtained from virtual cameras with random poses. As for the part segmentation dataset ShapeNetPart [28], 16,881 objects across 16 categories are available. And, for each object, ground truth part segmentation labels are provided. Note that both PCN and ShapeNetPart are subsets of a larger dataset ShapeNet [29]. To obtain point cloud objects with both partial observations and segmentation annotations, we consider the intersection objects of these two datasets, followed by normalizing/matching the ground truth point clouds for each object extracted. Then, the ground truth semantic labels (from ShapeNetPart) are assigned to each complete point cloud object and its partial versions. As a result, the intersection dataset contains 11,262 point cloud objects across 5 categories. Each object has 8 partial observations, while both complete and partial point clouds are with ground truth segmentation annotations. The above details can be seen in L2-12 of our supplementary materials and L208-214 in our main paper.
>
> **Q3: I noticed that you reproduced other methods for fair comparisons. I wonder if these methods are trained under the default configuration of the official implementation? Besides, both PMP-Net and PoinTr provide pre-trained models on PCN, how do these compare to your trained models?**
>
> A3: Yes, we implement and reproduce SOTA models under their default configurations, including learning rate, weight decay, etc. However, for fair comparison purposes, we train with one model for each category for all methods in our experiments, using the data explained in the above Q2. The above remarks are available in L223-225 in our main paper.

---

> > ### Author Response · Authors · 2022-08-02
> > **Response to Reviewer Bq58 (Part 2/2)**
> >
> > **Q4: Just to be sure, like other completion methods, SPoVT needs to train the model separately for each category of objects, and you can't mix parts of different categories of objects, right?**
> >
> > A4: We are glad to further clarify this issue. As explained in Q3, all methods including ours are trained for each object category for fair comparisons. However, with the suggestion from the reviewer, we now assess whether our SPoVT can be trained across different object categories. In fact, as stated in L97 of our main paper, we suggest that this can be achieved by increasing the total number of semantic parts M.
> >
> > We now conduct a new experiment that our SpoVT is trained on both “Airplane” and “Car” categories together in a unified model. The results are shown below: (CD: x$10^4$, mIoU: %)
> >
> >
> > |                 | Airplane |      |  Car |      |
> > |-----------------|:--------:|:----:|:----:|:----:|
> > |                 |    CD    | mIoU |  CD  | mIoU |
> > | Ours (Original) |   0.73   | 82.6 | 2.86 | 82.5 |
> > | Ours (Unified)  |   0.84   | 84.6 | 3.39 | 80.6 |
> >
> > From the above table, we observe that a unified SPoVT (trained for two object categories) slightly degraded the completion and segmentation performances when compared to the original SPoVT. Such degraded performances can be expected (due to the need to handle more object categories and parts in one model). Nevertheless, both models in the above table still performed favorably against SOTA methods listed in Table 2 (in which SOTA methods are trained for each category).
> >
> >
> >
> > **Q5: Some suggestions for figures. a) Figure 1 is too cluttered, lacks annotations, and has a different style for each block. This is confusing. b) In Figure 3, It might be better to mark the number of points in red. The writing of the paper needs to be improved a little bit. a) Some paragraphs are a little cluttered, like the Semantic VAE part, ablation studies part, etc. b) Typos, like “Tranformer-based” in the Introduction, “SWiVT” on Table 2, “mIOU”s , etc.**
> >
> > A5: We sincerely thank the reviewer for helping us improve the presentation. We will revise the figures and make them more readable, and the typo/errors will be corrected in the revised version (see below):
> >
> > * “Tranformer-based” in L47 -> “Transformer-based”
> > * “SWiVT” on Table 2 -> “SPoVT”
> > * “mIOU” on Table 2, Table 3,  L231, L235, L270 -> “mIoU”
> > * remove “Please refer to our supplementary for calculation details of T during the training process.”  in L181-182
> >
> >
> > **Q6: Please consider adding a discussion section to the paper to describe the limitations of the method as possible, such as the need for additional semantic annotations, inference speed, complex training process, etc.**
> >
> > We thank the reviewer for the valuable suggestion, and we feel that it is necessary to point out the limitations of our SPoVT below.
> >
> > As raised in Q1, our SPoVT requires ground truth semantic labels for the point cloud data during training, which might not be practically available. As discussed in Q1, we are able to alleviate this limitation by utilizing pre-trained segmentors for assigning point cloud labels.
> >
> > As for the concern about inference time, we do expect its increase when producing completion results with higher resolution. As stated in L195-201, since we produce such results by repeating the inference process multiple times, the inference time only grows linearly with the point cloud resolution (but not the memory usage). Please see the table below, in which we present the inference time and memory usage under different point cloud resolutions.
> >
> >
> > | Output point cloud resolution | Inference time (ms) | Memory usage (GB) |
> > |:-----------------------------:|:-------------------:|:-----------------:|
> > | 2048 points                   |         50.0        |       1.923       |
> > | 8192 points                   |        145.6        |       1.923       |
> > | 16384 points                  |        277.2        |       1.923       |
> >
> > Finally, training our SPoVT includes the pre-training of encoder and decoder for partial point cloud reconstruction, followed by completion and segmentation objectives. The details of our training process are presented in L185-194 in our main paper, which allows others to reproduce our model (if training his/her own models is of interest).
> >
> > We sincerely thank the reviewer again for the constructive suggestions, which make our work more complete and sound. We will add the above discussions in our revised version.

---

> > > ### Comment · Reviewer_Bq58 · 2022-08-08
> > > **Final Rating**
> > >
> > > Regarding my Q1, maybe I didn't express it clearly enough. The paper seems to directly use the evaluation of the segmentation results of DGCNN as a quality assessment for part generation. This evaluation method is somewhat reasonable, but not intuitive or common enough without explanation.
> > >
> > > However, after reading the authors' responses and the other reviewers' Q&As, almost all of my concerns have been addressed. I'll change my rating to 6: Weak Accept.

---

> > > > ### Author Response · Authors · 2022-08-09
> > > > **Follow-up comment**
> > > >
> > > > We sincerely thank the reviewer for willing to update the rating to weak accept (6) after the discussion period. We appreciate the reviewer for clarifying the above particular issue. We understand that the use of pre-trained DGCNN for evaluation and comparison may still be a concern. In recent works on semantic instance completion like [A, B], researchers take RGB-D images of indoor scenes as inputs, and they focus on producing completed 3D models with segmentation results for particular objects in that scene (e.g., chair, table, etc.). During the evaluation, these works also compare to methods particularly designed for scene completion or instance segmentation only. More specifically, when comparing to SOTAs for scene completion, they also apply pre-trained instance segmenter to produce the semantic labels for the completed outputs. In other words, the above evaluation protocol has been conducted (as what we did in Table 2). We will be happy to include the above discussions in the revised version, and we hope this would address the raised concern.
> > > >
> > > > [A] RfD-Net: Point Scene Understanding by Semantic Instance Reconstruction, CVPR 2021
> > > >
> > > > [B] RevealNet: Seeing Behind Objects in RGB-D Scans, CVPR 2020

---

### Meta-Review · Area_Chair_2aGp · 2022-08-31

**Recommendation:** Accept
**Confidence:** Certain

**Metareview:**

The paper received mixed reviews. After rebuttal, reviewers Bq58 and QsJ3 decided to raise the rating to weak accept. So all the reviewers give positive ratings and think the authors have addressed their concerns well. Taking the comments of the reviewers into account, the AC decided to accept this paper at NeurIPS.

**Award:**

No

---

### Decision · Program_Chairs · 2022-09-14

Accept